# Dynamic genome plasticity during unisexual reproduction in the human fungal pathogen *Cryptococcus deneoformans*

**Ci Fu**, **Aaliyah Davy**, **Simeon Holmes**, **Sheng Sun**, **Vikas Yadav**, **Asiya Gusa**, **Marco A. Coelho**, **Joseph Heitman***

Department of Molecular Genetics and Microbiology, Duke University Medical Center, Durham, North Carolina, United States of America

\* heitm001@duke.edu

**Data Availability Statement:** Nanopore and Illumina data for the XL280 genome have been deposited at the NCBI under the accession number PRJNA720102 that can be accessed through the

## Abstract

Genome copy number variation occurs during each mitotic and meiotic cycle and it is crucial for organisms to maintain their natural ploidy. Defects in ploidy transitions can lead to chromosome instability, which is a hallmark of cancer. Ploidy in the haploid human fungal pathogen *Cryptococcus neoformans* is exquisitely orchestrated and ranges from haploid to polyploid during sexual development and under various environmental and host conditions. However, the mechanisms controlling these ploidy transitions are largely unknown. During *C. deneoformans* (formerly *C. neoformans* var. *neoformans*, serotype D) unisexual reproduction, ploidy increases prior to the onset of meiosis, can be independent from cell-cell fusion and nuclear fusion, and likely occurs through an endoreplication pathway. To elucidate the molecular mechanisms underlying this ploidy transition, we identified twenty cell cycle-regulating genes encoding cyclins, cyclin-dependent kinases (CDK), and CDK regulators. We characterized four cyclin genes and two CDK regulator genes that were differentially expressed during unisexual reproduction and contributed to diploidization. To detect ploidy transition events, we generated a ploidy reporter, called *NURAT*, which can detect copy number increases via double selection for nourseothricin-resistant, uracil-prototrophic cells. Utilizing this ploidy reporter, we showed that ploidy transition from haploid to diploid can be detected during the early phases of unisexual reproduction. Interestingly, selection for the *NURAT* reporter revealed several instances of segmental aneuploidy of multiple chromosomes, which conferred azole resistance in some isolates. These findings provide further evidence of ploidy plasticity in fungi with significant biological and public health implications.

## Author summary

Ploidy is an intrinsic fundamental feature of all eukaryotic organisms, and ploidy variation and maintenance are critical to the organism survival and evolution. Fungi exhibit exquisite plasticity in ploidy variation in adaptation to various environmental stresses. For example, the haploid opportunistic human fungal pathogen *C. deneoformans* can generate diploid blastospores during unisexual reproduction and also forms polyploid titan cells

following link: https://www.ncbi.nlm.nih.gov/bioproject/PRJNA720102. Whole-genome Illumina data for the parental strains and the segmental aneuploid isolates have been deposited at the NCBI under the accession number PRJNA716858 that can be accessed through the following link: https://www.ncbi.nlm.nih.gov/bioproject/PRJNA716858. Underlying numerical data for all graphs and summary statistics were provided in S1 Data.

**Funding:** J.H. is supported by NIH/NIAID R37 award AI039115-24 and R01 awards AI050113-16 and AI133654-5. J.H. is codirector and fellow for the CIFAR program Fungal Kingdom: Threats & Opportunities program. The funders had no role in study design, data collection and analysis, decision to publish, or preparation of the manuscript.

**Competing interests:** The authors have declared that no competing interests exist.

during host infection; however, the mechanisms underlying these ploidy transitions are largely unknown. In this study, we elucidated the genetic regulatory circuitry governing ploidy duplication during *C. deneoformans* unisexual reproduction through the identification and characterization of cell cycle regulators that are differentially expressed during unisexual reproduction. We showed that four cyclin and two cyclin-dependent kinase regulator genes function in concert to orchestrate ploidy transition during unisexual reproduction. To trace and track ploidy transition events, we also generated a ploidy reporter and revealed the formation of segmental aneuploidy in addition to diploidization, illustrating the diverse mechanisms of genome plasticity in *C. deneoformans*.

## Introduction

Ploidy refers to the total number of chromosomal sets in a cell. Variations in ploidy are prevalent among both prokaryotic and eukaryotic organisms and have a profound effect on cellular phenotypes. Polyploidization has been suggested to provide adaptive advantages to environmental stresses through increases in gene copy number [1,2]. Cells can achieve polyploidization through either genome doubling within a single species, called auto-polyploidization, or via hybridization of genomes from different species, termed allo-polyploidization [2]. Upon polyploidization, cells experience the immediate impacts of having twice the genome content, which can include changes in cell size, genome stability, and gene expression. Despite these often drastic and deleterious changes, cells regularly tolerate ploidy transitions during mitotic and meiotic cell cycles, in which the entire genome undergoes duplication and reduction [1].

In the fungal kingdom, ploidy variation among natural isolates of a single species is a common phenomenon [3]. For example, the baker's yeast *Saccharomyces cerevisiae*, which is an evolutionary product of ancient allo-polyploidization between two different ancestral species, has natural isolates with ploidy ranging from haploid to tetraploid [3–6]. *Candida albicans*, which was once thought to be an obligate diploid human fungal pathogen, has been shown to form haploid, triploid, and tetraploid cells [7,8]. Nondiploid *C. albicans* cells have increased genomic instability and often return to a diploid or near-diploid state through auto-diploidization of the haploid genome or concerted chromosome loss of tetraploid cells, as *C. albicans* lacks a complete meiotic chromosomal reduction cycle [8,9]. In the syncytial hyphae of the filamentous fungus *Ashbya gossypii*, nuclear ploidy ranges from haploid to higher than tetraploid within the same hyphal compartment, and the degree of ploidy variation increases with hyphal aging and decreases upon exposure to cellular stress [10]. The prevalence of polyploidy in fungi illustrates how these genomic changes can provide efficient strategies for fungal cells to rapidly adapt to their environment [11].

Ploidy in the opportunistic human fungal pathogen *Cryptococcus* exhibits exquisite plasticity during sexual reproduction and under host infection conditions [12–14]. Cryptococcal infection can cause fatal cryptococcal meningitis in immunocompromised patients. The mortality rate of cryptococcal meningitis is as high as 70% for patients receiving treatment in resource-limited countries due to a lack of cost-effective therapeutics, and mortality is 100% in those left untreated [15]. *Cryptococcus* species have a bipolar mating system and undergo bisexual reproduction, while *C. deneoformans* can also undergo unisexual reproduction in the absence of a mating partner of the opposite mating type [16,17]. During bisexual reproduction, haploid *MAT*α and *MAT***a** cells undergo cell-cell fusion to achieve genome doubling, while during unisexual reproduction, haploid cells achieve genome doubling either via whole-genome duplication or through cell-cell fusion events between cells of the same mating type [18]. In the natural environment, *Cryptococcus* is largely present as haploid yeast cells, but

diploid cells of a single mating type (mainly αAAα) have also been documented, demonstrating that the presence of unisexual reproduction in nature can generate ploidy variation [19]. Besides ploidy transitions during sexual reproduction, *Cryptococcus* can also form polyploid giant cells, termed titan cells, during host infection [12,20]. The ploidy of titan cells can reach up to 64 or more copies of the genome, which is accompanied with morphological changes, including increased cell size up to 100 μm in diameter (compared to standard haploid cells that are 5 to 9 μm in diameter) and a thickened cell wall with a dense cross-linked capsule [21,22]. The formation of titan cells in host lung tissue has been shown to enable fungal evasion of phagocytosis by host immune cells and enhance fungal virulence [23,24]. Polyploid titan cells can further produce haploid and aneuploid progeny with enhanced tolerance to stressors within the host environment, and meiotic genes have also been shown to be activated in this niche [25,26]. To utilize this ploidy plasticity, *Cryptococcus* has evolved an elegant ploidy transition machinery that can be activated in response to mating cues, environmental stresses, and host conditions.

The environmental stimuli that trigger diploid and polyploid cell formation in *C. deneoformans* during unisexual reproduction and host infection have been characterized and include cell density and quorum sensing molecules, nutrient starvation, and serum [26–30]. However, the molecular mechanisms underlying these ploidy transitions are less clear. In other eukaryotic organisms, increases in ploidy are achieved primarily through endoreplication, during which, cells undergo multiple rounds of S phase without entering mitosis and cytokinesis [31,32]. This abnormal cell cycle is regulated by the same group of cyclins and cyclin-dependent kinases that govern the progression of the mitotic cell cycle [31,32]. For example, in flies and mammals, oscillation of cyclin E and cyclin-dependent kinase 2 activity is required for endocycles of S phase [31]. In *Schizosaccharomyces pombe*, mutants lacking the $P34^{cdc2}P56^{cdc13}$ mitotic B cyclin complex undergo multiple rounds of S phase and generate polyploid progeny [33]. In *S. cerevisiae*, cell cycle progression is regulated by activation of the cyclin-dependent kinase Cdc28 through binding of G1/S/G2/M-phase specific cyclins [34]. Periodical oscillation of B-type cyclin *CLB6* in *clb1-5Δ* cells can drive *S. cerevisiae* cells to re-enter S phase without undergoing mitosis and results in polyploid cell formation [35]. In *C. neoformans*, it was recently shown that reduced cyclin *CLN1* expression in cells arrested in G2 phase can lead to titan cell formation [36]. Thus, it is likely that concerted regulation of these cell cycle regulators in *C. deneoformans* contributes to diploidization during unisexual reproduction.

In this study, we sought to identify cell cycle regulators that govern ploidy transitions during unisexual reproduction in *C. deneoformans*. Because cell cycle progression in *S. cerevisiae* is governed by transcript levels of cyclins, we initially identified 20 putative cell cycle-regulating genes and examined their transcription levels during unisexual reproduction [34]. Among them, six genes were differentially expressed during unisexual reproduction compared to mitotic yeast growth. Further examination of the ploidy of blastospores, the diploid products of wild-type unisexual reproduction, confirmed that these genes are required for ploidy transitions during unisexual reproduction. We also developed a *NURAT* ploidy reporter to detect ploidy transition events and were able to detect both diploidization as well as aneuploid and segmental aneuploid formation events during both mitotic growth and unisexual reproduction, all of which underlie ploidy plasticity in *Cryptococcus* species.

## Results

### Identification of cell cycle regulators involved in unisexual reproduction

In fungi, ploidy duplication is a prerequisite for meiosis during sexual reproduction and is largely achieved through gamete fusion. However, cell fusion and nuclear fusion are

dispensable during unisexual reproduction in *C. deneoformans* and it has been proposed that an endoreplication pathway drives the haploid to diploid transition [18,37]. To elucidate the endoreplication pathway for unisexual reproduction, we sought to identify cell cycle regulators that are critical for this ploidy transition. Because cyclin abundance and turnover regulate cyclin-dependent kinase (CDK) activities and drive cell cycle progression [34], we searched for cyclins in the *C. deneoformans* JEC21 genome on FungiDB (www.fungidb.org) [38] and identified 51 candidate genes (S1 Table). Based on the annotated protein function for each gene, 20 genes were selected with predicted functions in the following three categories: cyclin (9), cyclin-dependent kinase (6), and CDK regulator (5) (S1 Table).

Transcriptional profiling during unisexual reproduction revealed that genes involved in the pheromone response pathway, meiosis, and spore production were activated between 24 and 48 hours upon mating induction [27,39]. We hypothesized that cell cycle genes important for ploidy duplication might be differentially expressed during unisexual reproduction. To determine this, we compared the expression levels of these putative cell cycle genes in wild-type *C. deneoformans* XL280α cells after incubation for 36 hours under mating-inducing conditions (V8 agar medium) to yeast cell growth conditions (nutrient-rich YPD medium) by qRT-PCR. *KAR5* served as a negative control as it was previously shown to be expressed at a comparable level under these two conditions [18]. We found four cyclin genes and two CDK regulator genes were significantly differentially expressed: *PCL2*, *CLB3*, and *CKS2* were downregulated, while *PCL6*, *PCL9*, and *CKS1* were upregulated (Fig 1A). Interestingly, none of the predicted cyclin-dependent kinase genes were differentially expressed (S1A Fig).

In agreement with our findings, all six genes were previously shown to be differentially expressed after growth on V8 medium for 12, 24, and 48 hours compared to growth on YPD for 12 hours (S1B Fig). *PCL6*, *PCL9*, and *CKS1* expression levels peaked at either the 24- or the 48-hour time point on V8 medium (S1B Fig). *CLB3* and *CKS2* were down-regulated on V8 medium, while *PCL2* had an initial upregulation on V8 at the 12-hour time point and then was down-regulated at later time points (S1B Fig) [39]. Three of the four cyclin genes, *PCL2*, *PCL6*, and *PCL9*, are Pho85 cyclins with predicted functions in regulating the cyclin-dependent kinase Pho85 in *S. cerevisiae* [40]. *CKS1* and *CKS2* are predicted to encode regulatory subunits for Cdc28, the master CDK for cell cycle progression in *S. cerevisiae* (S1 Table) [41].

To determine whether the differentially expressed cyclin and CDK regulator genes are required for unisexual reproduction, we generated two independent deletion mutants for each gene except for *CLB3*, for which we generated a galactose-inducible allele under the control of the *GAL7* promoter due to technical difficulty in deleting *CLB3* in our studies (S2A Fig). Deletion of *PCL2*, *PCL6*, *CKS1*, or suppressed expression of *CLB3* caused a mild reduction in hyphae production during unisexual reproduction, whereas deletion of *PCL9* or *CKS2* did not cause any defect in hyphae formation (Fig 1B). Deletion of *CKS1* or suppressed expression of *CLB3* also caused a defect in sporulation leading to the production of bald basidia (basidia lacking spores), strikingly different from the typical wild-type basidia with four chains of spores produced (Fig 1B). These results suggest *CKS1* and *CLB3* are required for the mitotic cycles during spore genesis, and they may also directly contribute to cell cycle progression during the meiotic cycle. The differential expression patterns and the observed morphological defects for the *pcl2Δ*, *pcl6Δ*, *cks1Δ*, and *P*$_{GAL7}$-*CLB3* strains suggest that these cell cycle regulatory genes play critical roles during *C. deneoformans* unisexual reproduction. Interestingly, deletion of these cell cycle regulating genes had a smaller impact on bisexual reproduction, especially that deletion of *CKS1* did not block basidium spore chain production (S2B Fig), further corroborating the hypothesis that the expression of these cell cycle regulating genes is coordinated during unisexual reproduction.

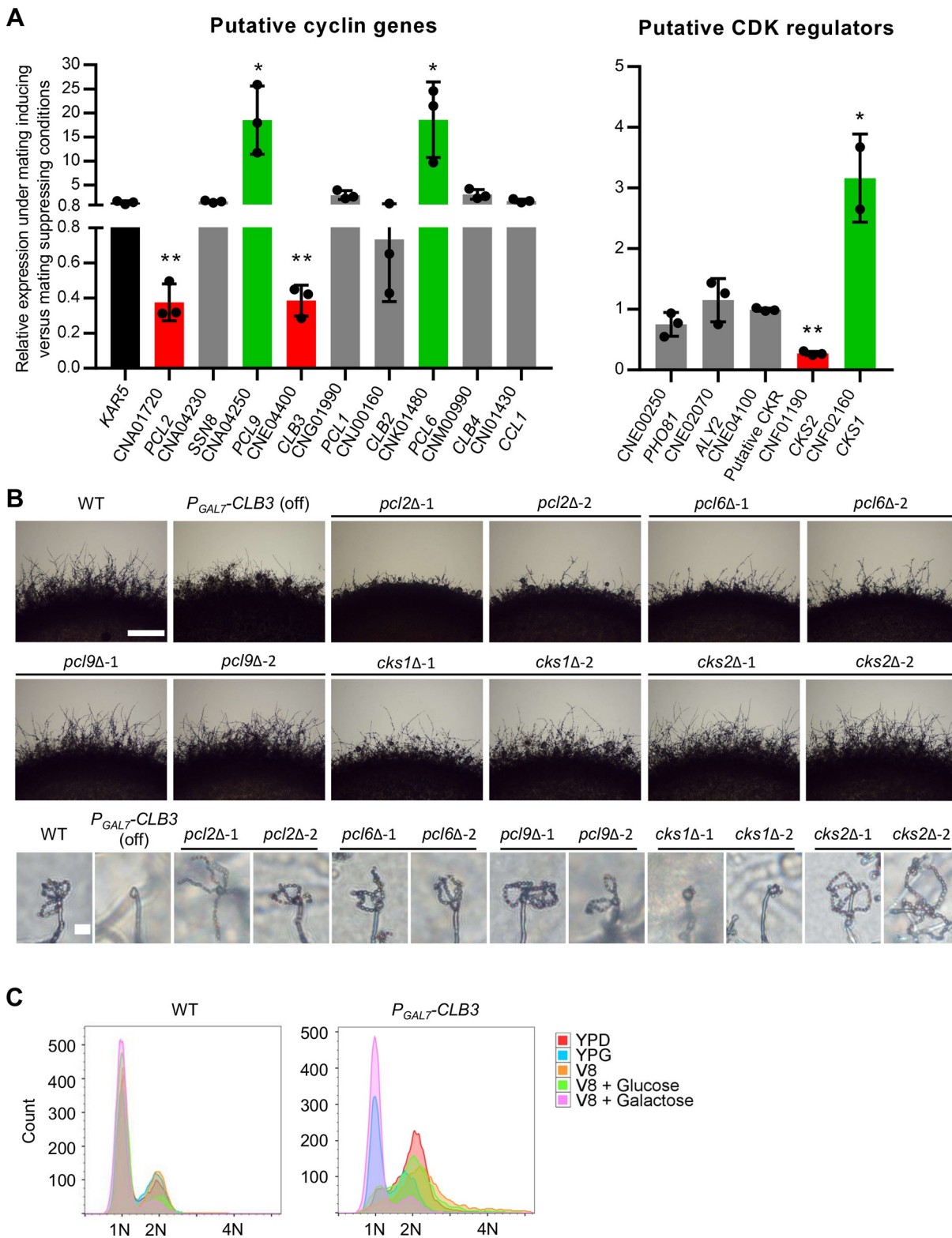

**Fig 1. Identification of cell cycle regulating genes involved in unisexual reproduction.** (A) Differential expression patterns of cell cycle regulating genes in wild type XL280α cells upon incubation for 36 hours on mating-inducing V8 agar medium versus nutrient rich YPD agar medium were examined by qRT-PCR. The error bars represent the standard deviation of the mean for three biological replicates. Red and green colors indicate genes that are significantly down- and up- regulated during unisexual reproduction compared to the control gene *KAR5*,

respectively. * indicates $0.01 < p \leq 0.05$ and ** indicates $0.001 < p \leq 0.01$. (B) Wild type XL280α, conditional expression mutant of *CLB3*, and deletion mutants of individual cell cycle regulating genes were grown on MS medium to assess unisexual hyphal growth and spore formation. Hyphal growth on the edge of each colony was imaged after 7 days and the scale bar represents 500 μm. Spore formation was imaged after three weeks and the scale bar represents 10 μm. (C) Wild type and the conditional expression strain for *CLB3* were grown on YPD, YPG, V8, V8 glucose, and V8 galactose for 24 hours. Ploidy for the cell populations were determined by FACS.

## *CKS1* and *CLB3* promote G2/M phase progression

To examine if these cell cycle regulating genes also function during yeast growth, we stained yeast cells with DAPI, which stains nuclei, and calcofluor white (CFW), which stains chitin in the cell wall, to observe yeast cell morphology for these mutant strains. Deletion of *CKS1* and suppressed expression of *CLB3* induced pseudohyphal growth whereas other deletion mutants or the expression of $P_{GAL7}$-*CLB3* in the presence of galactose all produced yeast cells with normal morphologies (S3 Fig), suggesting disruption in cell cycle progression can trigger pseudohyphal formation in *C. deneoformans*, similar to previous findings in *C. albicans* [42,43].

To determine if these genes are involved in cell cycle progression during yeast growth, we arrested cells from overnight cultures in liquid YPD medium at the G1/S phase with hydroxyurea (arrest was subsequently released by removing the reagent), and at G2/M phase with nocodazole [44–47]. Deletion of *CKS1* and suppressed expression of *CLB3* failed to respond to cell cycle arrest reagents and cells were arrested at G2/M phase in the overnight culture even before the treatment (S4 Fig and S4 Table), providing strong evidence that *CKS1* and *CLB3* promote G2/M phase progression during yeast growth. This cell cycle arrest may also contribute to the observed pseudohyphal growth in *cks1Δ* mutants and the $P_{GAL7}$-*CLB3* strain in the presence of glucose (S3 Fig).

## Cell cycle regulators contribute to ploidy duplication during unisexual reproduction

Cell cycle arrest at the G2 phase in large cell populations triggered by high temperature or nocodazole has been shown to promote hyphal growth in *C. deneoformans* [48], which is a hallmark of unisexual reproduction, illustrating a potential intrinsic association between ploidy transition and unisexual reproduction. To further characterize this association, we examined the ploidy distribution in populations of wild-type and mutant cells grown overnight on YPD and V8 agar media. Interestingly, although some mutants showed hyphal growth defects during unisexual reproduction (Fig 1B), all samples exhibited similar population distributions on mating-inducing medium compared to nutrient-rich YPD medium (S5 Fig). These results suggest that cell cycle arrest at G2 phase alone is not sufficient to promote unisexual reproduction, and that the ploidy transition required for unisexual meiosis likely occurs in a small portion of the cell population. Suppression of *CLB3* expression on V8 medium led to cell cycle arrest at the G2 phase and caused defects in both hyphal growth and sporulation (Fig 1B and 1C), further suggesting that cell cycle arrest at G2 phase is not sufficient to drive unisexual reproduction.

To understand how these cell cycle-regulating genes govern ploidy duplication, we examined the ploidy of blastospores, which are cells produced by mitotic budding directly from and along hyphae during unisexual reproduction. In the wild type, all blastospores tested from ten different budding sites of different hyphae were diploid except for two isolates that were aneuploid and originated from the same budding site (Fig 2 and S2 Table). These findings are in agreement with previous studies and provide evidence that diploidization occurs during or prior to blastospore formation [16,18]. Compared to the wild type, *pcl2Δ*, *pcl6Δ*, *cks1Δ*, and *cks2Δ* mutant strains produced blastospores with lower germination rates and suppressed

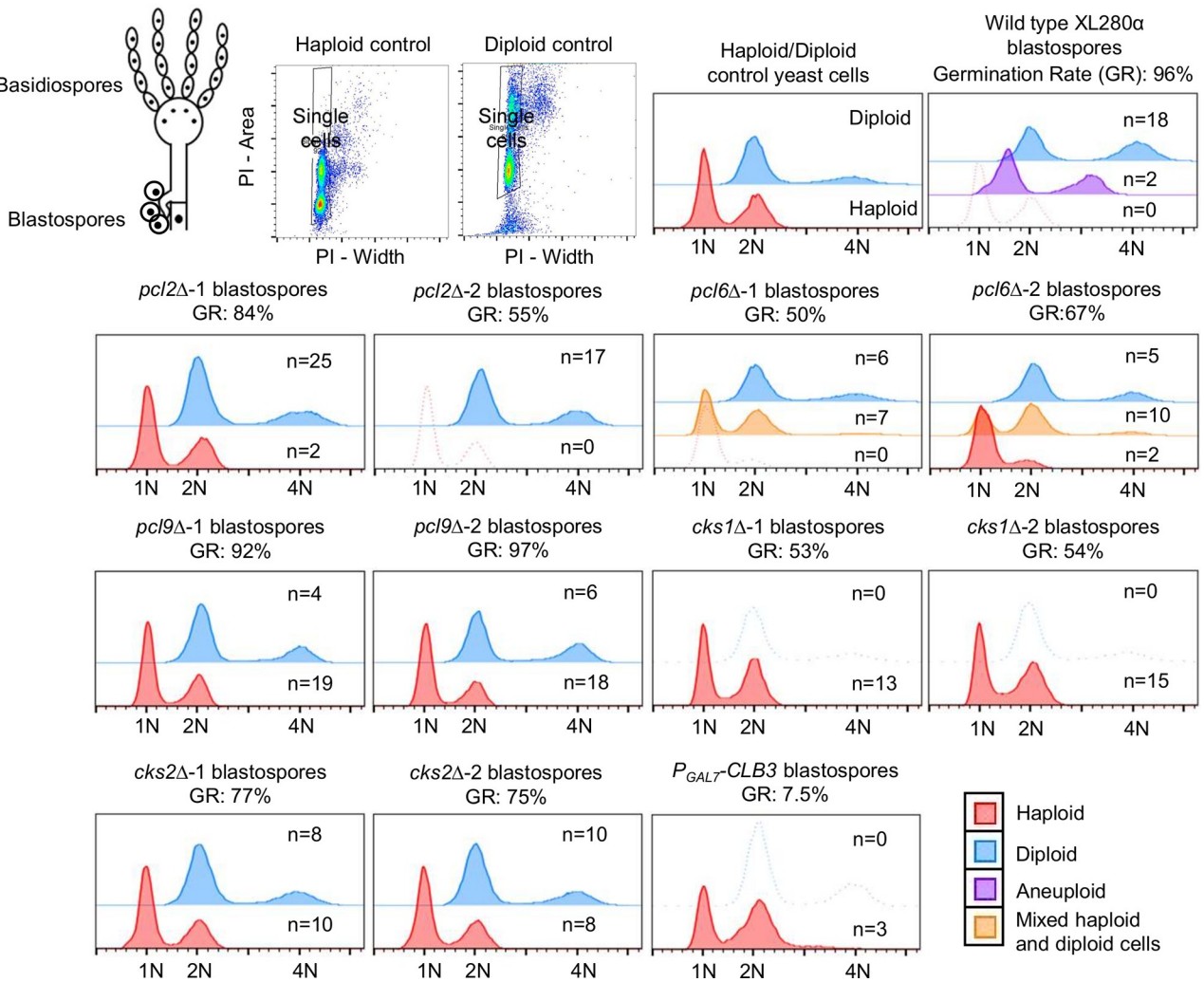

**Fig 2. Cell cycle regulating genes contribute to blastospore diploidization during unisexual reproduction.** Ploidy of single colonies derived from microscopically dissected blastospores were determined by FACS. Schematic diagram showing basidiospores and blastospores and representative gating strategy for single cells were provided at top left. Representative FACS results for haploid (red), diploid (blue), aneuploid (purple), or mixed haploid/diploid (yellow) were overlay-plotted with half offset.

expression of *CLB3* caused a severe defect in blastospore germination (S2 Table). Ploidy determination for these germinated blastospores showed that all of the gene deletions (except *PCL2*) as well as suppressed expression of *CLB3* caused a defect in diploid blastospore formation (Fig 2 and S2 Table). More than 50% of the *pcl6Δ* blastospores produced mixed populations of haploid and diploid cells, while about 80% of the *pcl9Δ* and 50% of the *cks2Δ* blastospores were haploid (Fig 2 and S2 Table). Deletion of *CKS1* and suppressed expression of *CLB3* led to the production of only haploid blastospores (Fig 2 and S2 Table). However, due to a defect in blastospore germination present in these mutants (S2 Table), we could not determine the ploidy status for these ungerminated blastospores. Collectively, our data suggest that these cell cycle regulating genes contribute to ploidy duplication during unisexual reproduction.

To further decipher the mixed haploid and diploid cell populations observed among the *pcl6Δ* blastospores and determine if *PCL6* is required for diploid maintenance, we streaked

cells derived from diploid wild-type, diploid *pcl6Δ*, and mixed haploid and diploid *pcl6Δ* blastospores for single colonies and subsequently determined their ploidy (S3 Table). Interestingly, all single colonies derived from diploid wild type and *pcl6Δ* blastospores were diploid, while all single colonies except one derived from mixed haploid-diploid *pcl6Δ* blastospores were either haploid or diploid (S3 Table). The observation of stable mitotically passaged diploid cells suggests that *PCL6* is not required for diploid maintenance during mitotic growth.

## Detecting ploidy transitions during unisexual reproduction with a ploidy sensor

Because the 1N and 2N population distributions largely remained the same during unisexual reproduction and under vegetative growth conditions in bulk culture (S5 Fig), we hypothesized that the ploidy duplication required for meiosis during unisexual reproduction might only be occurring in the sub-population of cells that are committed to unisexual hyphal growth. To track this hypothesized diploidization/endoreplication event, we engineered a genetic construct called *NURAT* that is similar to the *UAU1* cassette developed in *Candida albicans* [49] and allows for the detection of copy number increases in the genomic regions harboring this construct, which could be due to either aneuploidy formation or whole-genome duplication (Fig 3A). The *NURAT* construct encodes a functional *URA5* gene flanked by truncated 5' and 3' *NAT* cassette sequences that share 530 bp of the *NAT* coding sequence (CDS), which allows homologous recombination to yield a functional allele of the *NAT* cassette and thus conferring nourseothricin resistance (Fig 3A). We integrated the *NURAT* ploidy reporter into a previously identified safe haven locus on Chromosome (Chr) 1 in *MAT**a*** and *MATα* strains in which the native *URA5* gene had been replaced by the hygromycin resistance *HYG* cassette (S6 Fig) [50]. This ploidy reporter allows selection of nourseothricin resistant (NAT$^R$) and uracil-prototrophic (Ura$^+$) progeny; however, it depends on a copy number increase prior to homologous recombination in one of the two *NURAT* cassettes. If homologous recombination occurred between the truncated *NAT* CDSs in haploid cells prior to diploidization, the nourseothricin-sensitive (NAT$^S$) and uracil-prototrophic (Ura$^+$) haploid cell would become nourseothricin resistant (NAT$^R$) and prevent selection of the second copy of *NURAT* cassette due to the loss of the Ura$^+$ marker. Similarly, if homologous recombination occurred in both copies of the *NURAT* construct in diploidized cells, both *NURAT* cassettes become active *NAT* markers, which prevents the selection of NAT$^R$ Ura$^+$ diploid cells (Fig 3A). Normal homologous recombination functioning during either mitotic or meiotic growth is a prerequisite for the ploidy sensor to detect copy number variance.

To test the robustness of the *NURAT* reporter in detecting diploid cells versus haploid cells, we generated two diploid *NURAT/NURAT* strains through blastospore dissection of the haploid *NURAT* strain (S6 Fig) and performed fluctuation assays (S7 Fig). The haploid and diploid *NURAT* strains share the same genomic sequences and only differ in ploidy. In overnight liquid cultures, haploid and diploid *NURAT* strains exhibited similar *NURAT* recombination frequencies despite diploid *NURAT* strains having two copies of the *NURAT* construct (S8A Fig). Among the NAT$^R$ colonies, about 1% of the haploid *MAT**a*** cells and 0.04% of the *MATα* cells were Ura$^+$, whereas 73% and 79% of the diploid cells were Ura$^+$, suggesting homologous recombination occurred in only one copy of the *NURAT* construct in most diploid cells (S8A Fig). Interestingly, haploid and diploid NAT$^R$, Ura$^+$ colonies maintained their ploidy (S8A Fig). Our findings suggest that in the overnight cultures of haploid strains, very few cells undergo diploidization. Nevertheless, using the *NURAT* construct, we were still able to detect possible aneuploidy of chromosome 1 (on which the *NURAT* construct is located) that occurred at a low frequency.

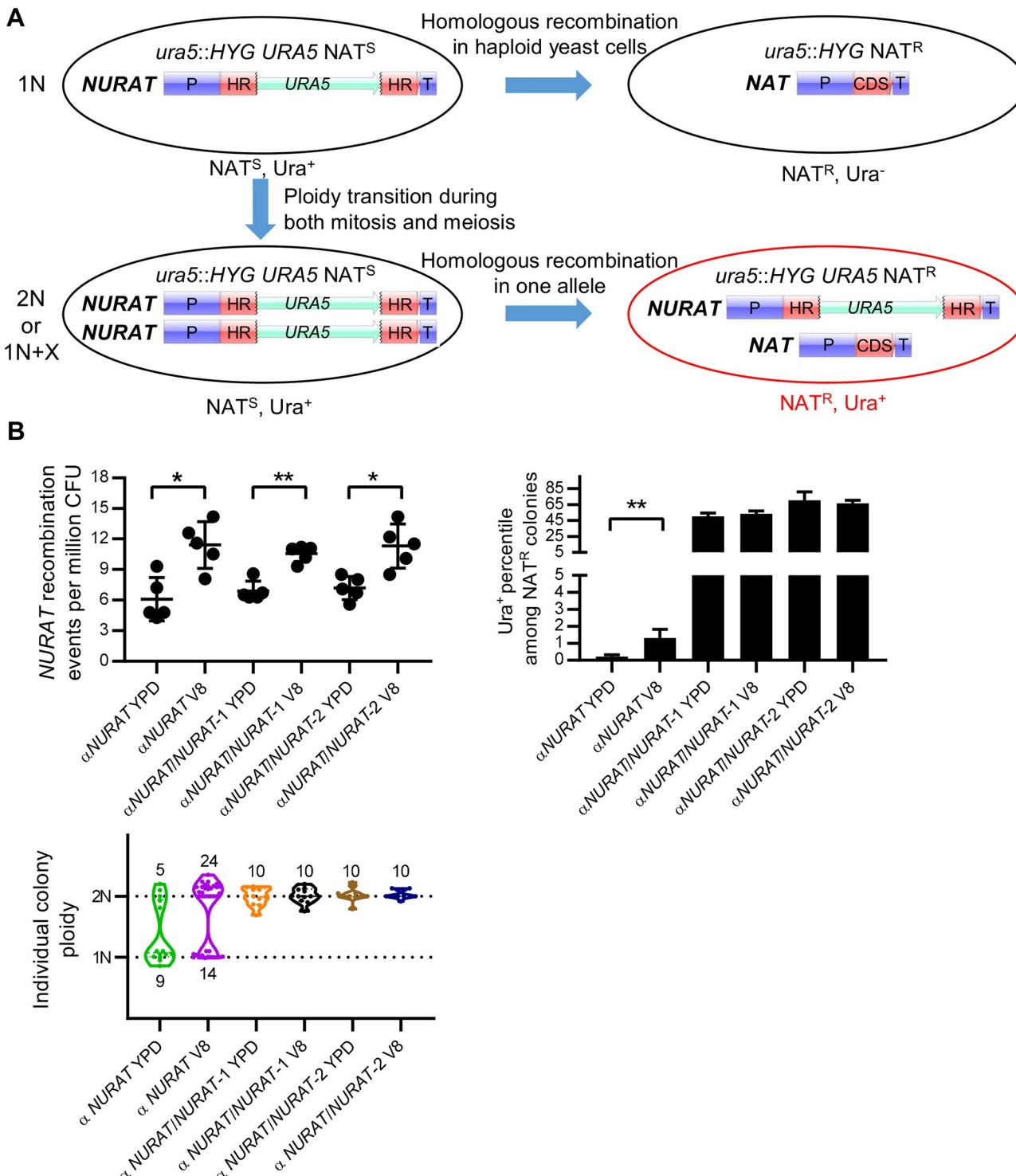

**Fig 3. Ploidy sensor reports ploidy transition events during both mitotic and meiotic growth.** (A) Schematic diagram shows that the mechanism by which the ploidy sensor construct *NURAT* detects ploidy transition depends on homologous recombination in one *NURAT* allele after the ploidy transition events. (B) A haploid and two diploid *ura5Δ* strains harboring *NURAT* constructs were incubated on YPD and V8 agar medium for 60 hours. The number of *NURAT* recombination events per million CFU was plotted to compare the recombination frequencies. NAT[R] colonies were replica-plated onto SD-URA agar medium to obtain cells that contain both *NURAT* and *NAT* constructs. The percentile of *NURAT* presence among NAT[R] colonies were plotted in the bar graph. Mean values of five independent experiments were plotted for the *NURAT* recombination frequencies and the Ura[+] percentiles among NAT[R] colonies; error bars represent standard deviations. * indicates $0.01 < p \leq 0.05$ and ** indicates $0.001 < p \leq 0.01$. Individual *NAT/NURAT* colonies were tested for ploidy by FACS and plotted in the violin plot.

To examine whether the *NURAT* reporter could be used to detect ploidy transition events during unisexual reproduction, we incubated haploid and diploid *MAT*α *NURAT* strains on both YPD and V8 agar media for 36 and 60 hours (S7 Fig). After 36 hours of incubation on the mating-inducing V8 medium, only haploid *NURAT* cells displayed a significant increase in *NURAT* recombination compared to cells incubated on YPD (S8B Fig). Interestingly, both haploid and diploid cells showed a significant increase in *NURAT* recombination after incubation for 60 hours on V8 medium (Fig 3B), illustrating a possible elevated rate in homologous recombination under mating-inducing conditions. Among the recombined NAT$^R$ colonies, haploid cells had a much lower percentage of Ura$^+$ colonies compared to diploid cells (about 1% in haploid and 50–70% in diploid) (Fig 3B). Additionally, activation of unisexual reproduction after longer incubation on V8 medium did not lead to increased numbers of NAT$^R$, Ura$^+$ cells in the diploid populations, but a significant increase of NAT$^R$, Ura$^+$ cells was observed in the haploid populations (Figs 3B and S8). FACS analyses of individual colonies showed that there were more diploid colonies than haploid cells after longer incubation on V8 (Figs 3B and S8), suggesting that diploidization occurs during unisexual reproduction, and the *NURAT* construct can indeed detect ploidy transition events. However, the low frequency of detected diploidization events in this assay also suggests that ploidy duplication during early time points of unisexual reproduction is occurring in a sub-population of cells whereas other cells within the mating patch undergo mitotic growth as haploid isolates. It is important to note that the sensitivity of this ploidy sensor to detect ploidy changes is limited by the frequency of mitotic recombination of the *NURAT* reporter.

## Segmental aneuploidy occurs during both mitosis and meiosis

To understand the nature of the NAT$^R$, Ura$^+$ colonies, which harbor both *NAT* and *NURAT* alleles (referred to subsequently as *NURAT/NAT* strains), we performed whole-genome Illumina sequencing of five *NURAT/NAT* colonies derived from mitotic passages of both the *MAT***a** and *MAT*α *NURAT* strains (S6 Fig). Chromosome alterations were inferred from changes in coverage of reads aligned to a newly obtained, chromosome-level, reference genome assembly of *C. deneoformans* XL280α, generated *de novo* by Oxford Nanopore sequencing (see Materials and Methods for details). Instead of observing the expected aneuploidy of Chr 1 where the *NURAT* construct has been inserted (the safe haven locus is located near the end of the chromosome arm), segmental aneuploidy and about one extra copy of the region harboring the *NURAT* construct was present in all of the *NURAT/NAT* colonies (Figs 4A, 4B and S9). Besides the segmentally duplicated region on Chr 1 that was selected for, other chromosomal abnormalities were also detected, including segmental duplications of Chr 2 in the diploid *MAT***a** *ura5Δ NURAT/NAT*-4 isolate, Chr 6 in the *MAT*α *ura5Δ NURAT/NAT*-1 isolate, Chr 10 in *MAT***a** *ura5Δ NURAT/NAT*-2 isolate, and Chr 13 in the *MAT***a** *ura5Δ NURAT/NAT*-1 and -2 isolates (Fig 4A). Interestingly the segmentally duplicated region on Chr 13 in *MAT***a** *NURAT/NAT* colonies 1 and 2 spanned the centromere, which could potentially give rise to a dicentric chromosome. Loss of chromosomal segments (inferred as regions with lower read coverage) were only detected in the context of the diploid *MAT***a** *ura5Δ NURAT/NAT*-4 isolate (on Chrs 2 and 4).

Segmentally duplicated regions on Chr 1 in all progeny also harbored a drug efflux pump gene, *AFR1*, which has been shown to confer fluconazole resistance in *Cryptococcus* (Fig 4B) [51]. Interestingly, three out of five strains tested were found to be resistant to fluconazole, suggesting additional epistatic interactions involving the *AFR1* gene or unidentified mutations mitigated the fluconazole resistance phenotype conferred by *AFR1* gene copy number increase (Fig 4C). We also found that two of the strains were hypersensitive to 37˚C, a phenotype that has been associated with aneuploidy in *C. neoformans* (Fig 4C) [14].

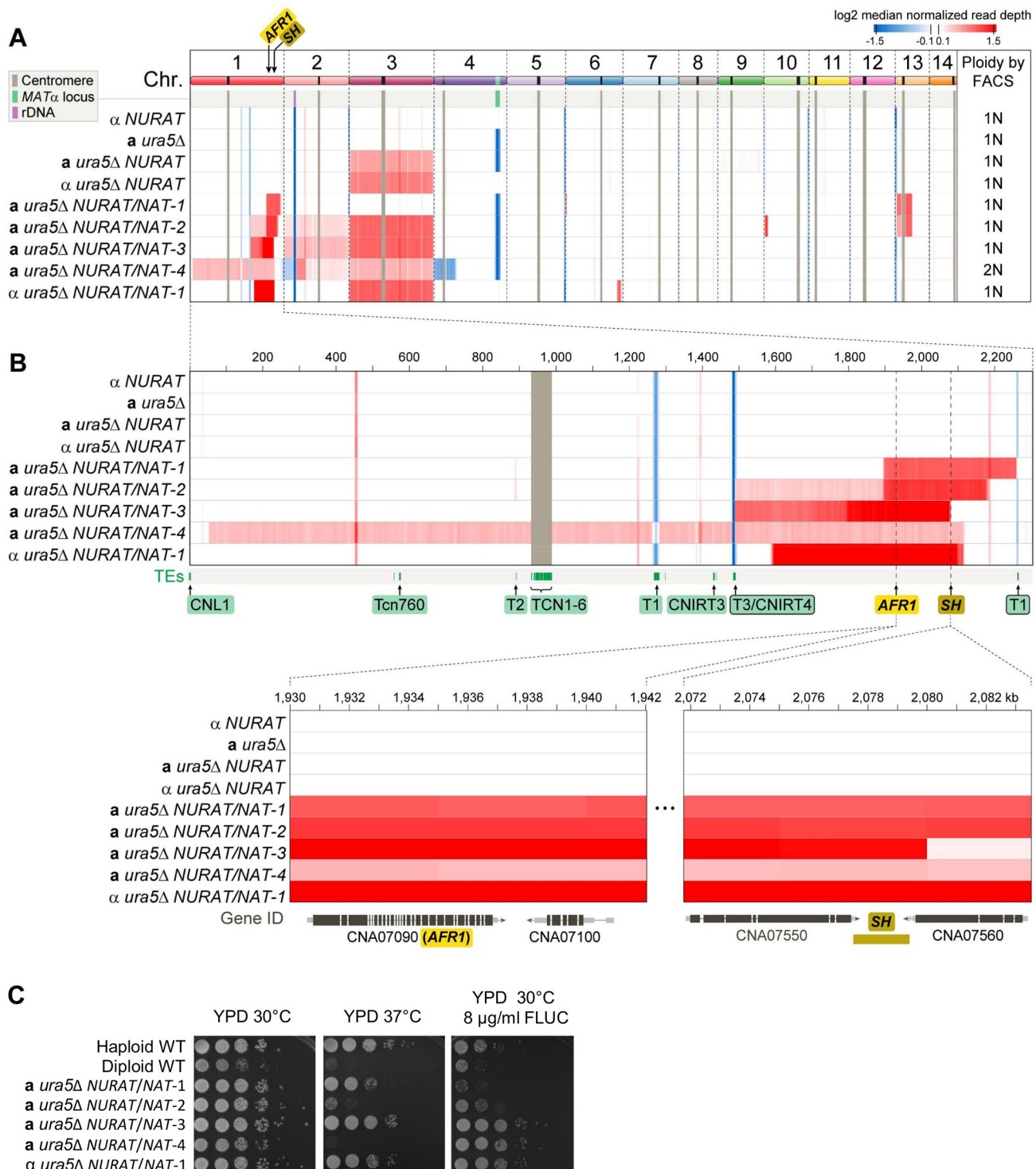

**Fig 4. Ploidy sensor detects auto-diploidization and segmental aneuploid formation.** (A) *MATα NURAT*, *MAT**a** ura5Δ*, *MAT**a** ura5Δ NURAT*, *MATα ura5Δ NURAT*, and four *MAT**a** ura5Δ NURAT/NAT* and one *MATα ura5Δ NURAT/NAT* colonies derived through mitotic passaging were subjected to Illumina whole-genome sequencing. Read depth across all 14 chromosomes was plotted for each strain. Ploidy was determined by FACS and is listed at the end of each sequencing result. Centromeres and mating-type loci are indicated by grey and green bars, respectively. (B) Read depth for Chr 1 and regions containing the *AFR1* gene, which encodes a drug efflux pump, and the safe haven locus, where *NURAT* is inserted. Both loci are present in duplicated chromosomal segments among all *NURAT/NAT* progeny. Transposable elements were highlighted and labeled in green below the Chr 1 read depth plot. (C) Human host temperature tolerance and fluconazole resistance phenotypes were examined for these five *NAT/NURAT* strains with haploid and diploid wild-type controls. Cells were 10-fold serial-diluted and spotted on YPD and YPD supplemented with 8 µg/ml fluconazole, and then incubated at either 30°C or 37°C for two days.

## Segmental aneuploidies are formed via multiple mechanisms

To elucidate the mechanism(s) giving rise to segmental aneuploidy, we analyzed the genomic regions flanking the segmental duplications detected in different *NURAT/NAT* progeny. Specifically, we assessed if read-pairs aligned to those regions in the XL280α reference genome had unexpected separation distances, anomalous orientations, or if the forward and reverse reads of a pair aligned to different chromosomes, all indicative of structural changes. This analysis revealed distinct modes of segmental aneuploid formation among different *NURAT/NAT* progeny (S9 Fig). Sequencing reads bridging segmentally duplicated regions on different chromosomes were detected in three isolates suggesting potential fusion events may have occurred between segmentally duplicated regions: Chrs 1 and 13 (a1-a2) and Chrs 1 and 6 (b1-b2) in the *MAT**a** ura5Δ NURAT/NAT*-1 strain, Chrs 1 and 13 (a1-a2 and c1-c2) and Chrs 1 and 10 (b1-b2) in the *MAT**a** ura5Δ NURAT/NAT*-2 strain, and Chrs 1 and 2 (a1-a2) and Chrs 2 and 4 (c1-c2) in the *MAT**a** ura5Δ NURAT/NAT*-4 strain (S9 Fig). In all three isolates, one segmentally duplicated region encompasses the centromere (Chr 13 in the *MAT**a** ura5Δ NURAT/NAT*-1 and -2, and Chr 1 in *MAT**a** ura5Δ NURAT/NAT*-4) allowing the opportunity for neo-chromosome formation through fusion of segmentally duplicated regions originated from different chromosomes. Segmental aneuploidy formation via tandem duplications were detected for Chr 1 in the *MAT**a** ura5Δ NURAT/NAT*-3 strain and for Chrs 1 and 6 in the *MAT*α *ura5Δ NURAT/NAT*-1 strain, in which large and small inversion events were detected, suggesting that complex chromosomal rearrangements are also associated with segmental aneuploidy formation. Transposable element movements have been shown to be highly mutagenic, especially under host infection or temperature stress [52]. Interestingly, T1 and T3/CNIRT4 transposon movements were detected in sequences flanking some of the *NURAT/NAT* progeny, suggesting that transposable elements may have also contributed to the formation of some of the segmental aneuploidies (S9 Fig).

The segmentally duplicated regions were further analyzed through separation of chromosomes via CHEF gel electrophoresis followed by chromo-blotting with probes specific to these regions. These methods revealed that various forms of karyotypic changes were present in the *NURAT/NAT* progeny (S10 Fig). The *NAT* probe for the *NURAT* construct in the *MAT**a** ura5Δ NURAT/NAT*-1 strain and both the *NAT* and *URA5* probes in the *MAT**a** ura5Δ NURAT/NAT*-2 strain hybridized to a smaller chromosome than expected (S10A Fig, green arrows). Additionally, the probe specific to the duplicated region on Chr 13 in the *MAT**a** ura5Δ NURAT/NAT*-1 strain hybridized to two smaller chromosomes, supporting the hypothesis that segmentally duplicated regions can form neochromosomes (S10C Fig, green arrows). Hybridization of the *NAT* and *URA5* probes to a smaller chromosome in the *MAT**a** ura5Δ NURAT/NAT*-3 and the *MAT*α *ura5Δ NURAT/NAT*-1 strains suggested the segmentally duplicated events "likely did not" occur directly on Chr 1 (S10A Fig, red arrows). Conversely, hybridization of a probe specific to the segmentally duplicated region on Chr 6 in the *MAT*α *ura5Δ NURAT/NAT*-1 strain supported the tandem duplication of the region within Chr 6, as detected by whole-genome sequencing (S9 and S10B Figs, green arrow). Overall, the different types of segmental aneuploidy formation illustrate a significant level of genomic and ploidy plasticity in *C. deneoformans*.

## Cell cycle regulators contribute to segmental aneuploidy formation

To investigate how the identified cell cycle-regulating genes impact ploidy transitions during unisexual reproduction, we generated mutant strains containing both the *NURAT* construct and the *ura5* gene deletion through meiotic crosses and performed fluctuation assays (S7 Fig). Compared to the wild type, genetic deletion of *PCL2*, *PCL6*, *PCL9*, *CKS1*, and *CKS2* all

significantly reduced *NURAT* recombination frequencies. *pcl6Δ* and *cks1Δ* mutants exhibited the most severe defects, suggesting modulation of ploidy transitions by these cell cycle-regulating genes could influence the regulation of homologous recombination during unisexual reproduction (Fig 5A). Interestingly, suppressed expression of *CLB3* significantly increased *NURAT* recombination whereas upregulated expression of *CLB3* (by supplementing galactose in V8 medium) significantly reduced *NURAT* recombination, indicating that *CLB3* plays an opposite role in contributing to homologous recombination frequencies during unisexual reproduction. Addition of glucose or galactose in V8 medium also increased homologous recombination frequencies in the wild type, which is likely due to robust vegetative growth in the presence of excess nutrients.

Despite the significantly reduced *NURAT* recombination frequencies observed in the deletion mutants, they produced significantly different frequencies of NAT$^R$, Ura$^+$ colonies than the wild type except for *pcl9Δ* mutants. *pcl2Δ*, *pcl6Δ*, and *cks2Δ* mutants produced zero or only one NAT$^R$, Ura$^+$ colony (Fig 5A), suggesting that these three genes all function in driving diploidization and segmental aneuploidy formation during unisexual reproduction. In contrast, *cks1Δ* mutants, which had a severe defect in *NURAT* recombination, produced significantly more NAT$^R$, Ura$^+$ colonies than the wild type (56% in *cks1Δ*-1 and 32% in *cks1Δ*-2 compared to 0.9% in wild type) (Fig 5A), indicating that *CKS1* plays an inhibitory role in diploidization and segmental aneuploidy formation. Interestingly, both suppressed and upregulated expression of *CLB3* increased the frequency of NAT$^R$, Ura$^+$ colonies relative to the wild type (Fig 5B), suggesting that *CLB3* expression levels modulate diploidization and segmental aneuploid formation.

To evaluate the impact of these cell cycle genes on ploidy transitions, the average ploidy was calculated for all *NURAT*/*NAT* colonies of wild type and mutants (Fig 5). For *pcl9Δ* mutants that produced a comparable number of NAT$^R$, Ura$^+$ colonies to wild type, deletion of *PCL9* significantly increased the average ploidy level when outliers with ploidies above 1.6 were removed from the dataset (S11A Fig). Both suppression and upregulation of *CLB3* expression significantly decreased the average ploidy of the *NURAT*/*NAT* colonies when the same outliers were removed (S11B Fig). Taken together, our findings suggest that cell cycle-regulating genes act in concert to control ploidy transitions during unisexual reproduction.

## Discussion

Genome size changes occur during both mitotic and meiotic cycles and disruption of these processes leads to dire consequences on cellular viability and fertility. During mitosis, the whole genome duplicates during S phase and is governed by cell cycle regulators, while in fungi, during meiosis, ploidy duplication is accomplished via cell-cell fusion and nuclear fusion between mating partners [1,32]. Besides these two fundamental cellular processes, ploidy transitions also occur during unisexual reproduction and titan cell formation in the human fungal pathogen *C. deneoformans*, in which ploidy increases through putative endoreplication pathways [13,18,21,36]. To examine the mechanism(s) underlying the ploidy transition from haploid to diploid during unisexual reproduction, we identified and characterized six cell cycle-regulating genes that contribute to diploidization.

Among the identified twenty putative cell cycle-regulating genes, four cyclin genes and two CDK regulator genes were differentially expressed in *C. deneoformans* during unisexual reproduction compared to mitotic growth, whereas none of the cyclin-dependent kinase genes exhibited expression differences. This finding is in agreement with previous studies in *S. cerevisiae* that found fluctuations in transcript levels of cyclin genes, but not CDK genes, drive cell cycle progression [34]. Three of the four cyclin genes, *PCL2*, *PCL6*, and *PCL9*, are Pho85

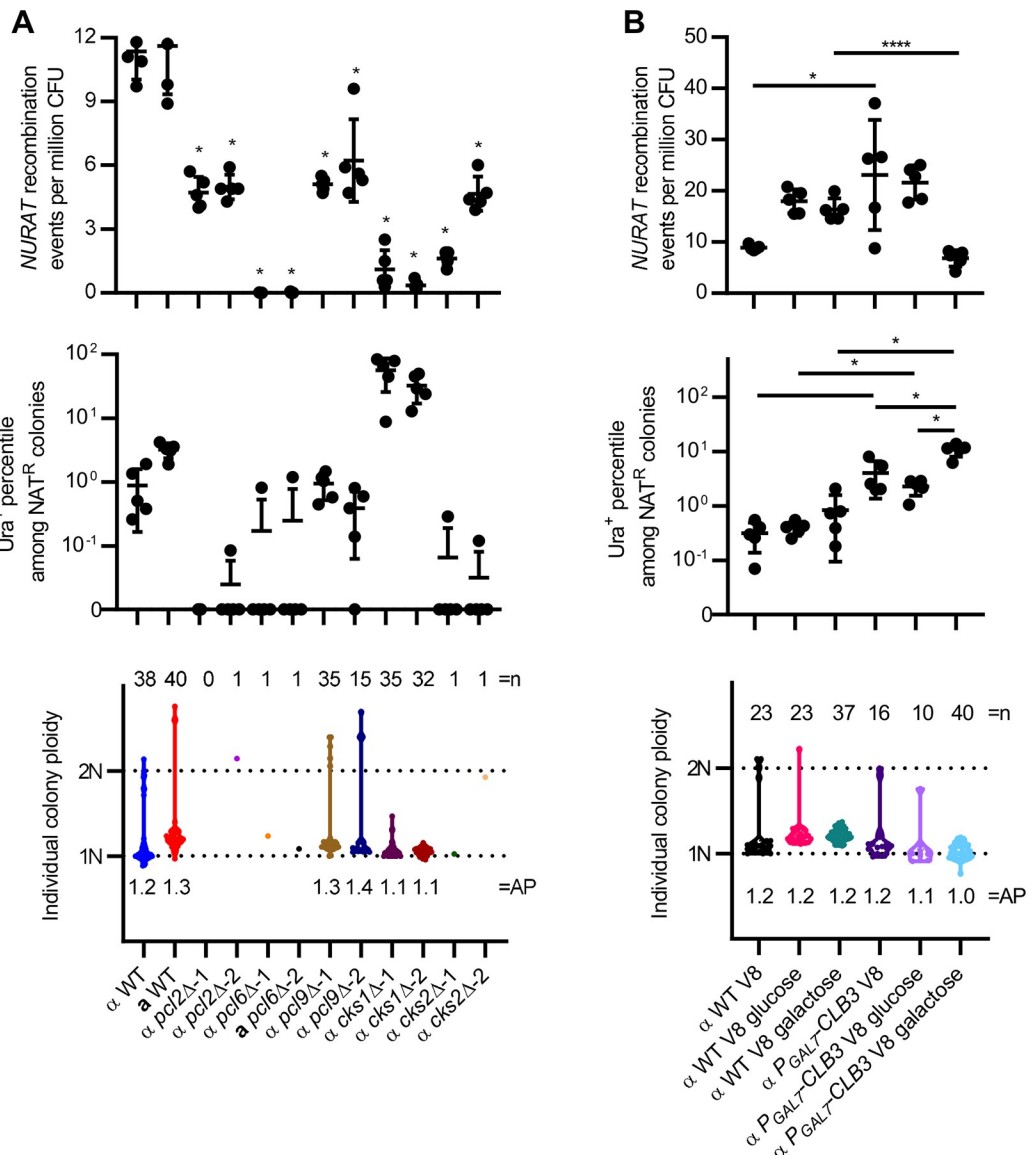

**Fig 5. Cell cycle regulating genes contribute to segmental aneuploid formation.** (A) Deletion of *PCL2*, *PCL6*, *PCL9*, *CKS1*, and *CKS2* reduced *NURAT* recombination frequency; deletion of *CKS1* increased the percentile of *NURAT* presence among NAT[R] colonies; deletion of *PCL2*, *PCL6*, and *CKS2* blocked segmental aneuploidy formation. Student's t-test with Bonferroni correction for 10 repeated tests was performed to compare each mutant with the wild type. *p* value lower than 0.005 (*) was considered statistically significant. (B) Suppressed expression of *CLB3* increased *NURAT* recombination frequency and the percentile of *NURAT* presence among NAT[R] colonies, while increased expression of *CLB3* reduced *NURAT* recombination frequencies but increased the percentile of *NURAT* presence among NAT[R] colonies. Neither down regulation or up regulation of *CLB3* blocked segmental aneuploid formation. Mean values of five independent experiments were plotted for the *NURAT* recombination frequencies and the Ura+ percentiles among NAT[R] colonies; error bars represent standard deviations. Student's t-test with Bonferroni correction for 9 repeated tests was performed for each pairwise comparison. *p* value lower than 0.0056 (*) was considered statistically significant.

cyclins (Pcls), which in *S. cerevisiae* interact with the cyclin-dependent kinase Pho85 [53]. Pcl2 and Pcl9 belong to the Pcl1,2-like subfamily and are expressed during G1 phase activating Pho85, which promotes the G1/S phase transition [40,53]. Pcl6 belongs to the Pho80 subfamily and functions with Pho85 in activating the serine/threonine protein phosphatase Glc7, which modulates kinetochore-microtubule interactions during M phase [53]. In *C. deneoformans*,

expression of *PCL6* and *PCL9* was highly induced during unisexual reproduction, and their deletion caused a defect in diploid blastospore formation, while in contrast, *PCL2* was down-regulated during unisexual reproduction and its deletion did not cause a defect in diploid blastospore formation, suggesting concerted regulation of these cyclins are critical in driving diploidization during unisexual reproduction, likely via modulation of Pho85 activity. The fourth identified gene, *CLB3*, belongs to B-type cyclins and its expression was reduced during unisexual reproduction. However, repression of *CLB3* during unisexual reproduction blocked diploid blastospore formation. In *S. cerevisiae*, *CLB3* is a nonessential cyclin gene expressed during G2/M phase, activating Cdc28 and promoting G2/M phase transition [34]. In *C. deneoformans*, *CLB3* was also not essential, and Pengjie Hu and colleagues have successfully deleted *CLB3* utilizing a CRISPR-mediated transformation technique [54,55]. Our failure in deleting this gene was likely due to the combinatory effect of low biolistic transformation efficiency and a severe defect in cytokinesis of the mutant. Deletion of *CLB3* caused defects in stress resistance, melanin production, and capsule formation, and the deletion strain was avirulent in a mouse infection model [54,56]. Interestingly, different from the deletion mutants of the other five cell cycle regulating genes, deletion of *CLB3* blocked basidiospore chain production during bisexual reproduction [54], suggesting that Clb3 plays a pivot role in sporogenesis during both modes of sexual reproduction in *C. deneoformans*. In this study, suppression of *CLB3* expression led to cell cycle arrest at G2/M phase, which is consistent with the phenotype observed for the deletion mutant, suggesting that Clb3 contributes to G2/M phase progression, a conserved role for Clb3 in *S. cerevisiae* [34,54].

The two differentially expressed CDK regulator genes *CKS1* and *CKS2* are both homologs of *S. cerevisiae CKS1*, which is required for G1/S and G2/M phase transitions [41]. In *S. cerevisiae*, Cks1 functions as a phosphor-adaptor protein for the CDK inhibitor Sic1 and the G1/S phase Cln-Cdc28 complex, which facilitates phosphorylation and destruction of Sic1 at the onset of S phase [57]. Similar to *C. deneoformans*, humans have two homologs of the *S. cerevisiae CKS1* gene, and both can complement a null mutation of *CKS1* in *S. cerevisiae* [58]. Interestingly, the *C. deneoformans CKS1* and *CKS2* genes were oppositely regulated during unisexual reproduction. Deletion of *CKS1*, which is upregulated during unisexual reproduction, caused a more severe phenotype during unisexual spore production and diploid blastospore formation compared to deletion of *CKS2*, which is downregulated during unisexual reproduction. These results suggest a functional divergence between *CKS1* and *CKS2* in *C. deneoformans*.

Among the six characterized cyclin and CDK regulator genes, *PCL9*, *CLB3*, and *CKS2* were previously shown to be periodically expressed during the cell cycle in *C. neoformans* with a peak at G2/M phase for *PCL9* and a peak at G1/S phase for *CLB3* and *CKS2* [59]. Although *PCL2* and *CKS1* were not periodically expressed during the cell cycle, expression of both genes peaked at G1/M phase during the first cell cycle in synchronized cells in *C. neoformans*, supporting a role for these cell cycle regulating genes in cell cycle progression [59]. Four of these cell cycle-regulating genes (*CLB3*, *PCL2*, *CKS1*, and *CKS2*) were highly expressed in the G1/S phase under rich growth conditions in *C. neoformans*; however, deletion of *CKS1* and repression of *CLB3* in *C. deneoformans* caused a prominent phenotype in G2/M phase arrest. More interestingly, *CKS1* and *CLB3* expression were differentially regulated during unisexual reproduction, suggesting a rewired transcriptional coordination of different cell cycle-regulating genes may be required for genome duplication and diploid genome maintenance before the onset of meiosis during unisexual reproduction. However, mechanisms underlying this process remain to be elucidated.

Different from unisexual reproduction, ploidy increases during titan cell formation was recently shown to be modulated by *CLN1* expression in *C. neoformans* [36]. Interestingly, the

orthologous *CLN1* gene (CNM00990) in *C. deneoformans* was not differentially expressed during unisexual reproduction (Fig 1A). Under host environment, unbudded *C. neoformans* cells were arrested in G2 with a 2C genome and reduced *CLN1* expression allowed 2C cells to re-enter the G1/S phase without mitosis to form polyploid titan cells [36] These distinct cyclin expression regulations during different cellular and developmental processes highlight the exquisite genome and ploidy adaptability of *Cryptococcus* species in response to various host and environmental stress cues.

To further detect ploidy transition events during unisexual reproduction, we generated the *NURAT* ploidy reporter, using which, we were able to detect diploid cells under both mating-suppressive (YPD) and mating-inducing (V8) culture conditions by selecting NAT$^R$, Ura$^+$ colonies. Although more diploid cells were detected on V8 compared to YPD, the overall frequency of diploid cells was low, suggesting that diploidization may occur in only a small number of cells that are primed for unisexual reproduction, that diploidization events could be transient, or that the relative sensitivity of detection with the *NURAT* reporter is limited by the frequency of recombination.

Among the selected NAT$^R$, Ura$^+$ colonies from both YPD and V8 culture condition, many remained haploid. Whole-genome sequencing of these haploid colonies showed that these colonies had segmental aneuploidies of the region on Chr 1 where the *NURAT* cassette was inserted. Besides the selected region on Chr 1, other chromosomes which were not under selection also exhibited segmental aneuploidy formation. Segmental aneuploidy represents a form of chromosome instability, a hallmark of tumorigenesis, which occurs via breakage-fusion-bridge cycles of duplication and multiple amplifications of certain chromosome regions [60]. In the human fungal pathogen *C. albicans*, segmental aneuploidy has been observed in azole-resistant isolates derived clinically and experimentally, with multiple amplified regions containing genes, such as *TAC1* and *ERG11* that contribute to azole tolerance [61–63]. In *C. albicans*, segmental aneuploidy is observed exclusively in regions flanked by long inverted repeat sequences, which function in repairing the breakage-fusion-bridge cycles during the formation of segmental aneuploidy [63,64]. Long-repeat sequences are distributed throughout the *C. albicans* genome, suggesting a strong potential for segmental aneuploidy formation [63]. Interestingly, in the absence of the drug, the azole-resistant, segmentally aneuploid isolates can quickly lose the multiple amplified copies and return to euploid chromosomal karyotypes and azole susceptibility, suggesting a selective pressure may be required for the maintenance of segmental aneuploidy [64].

Different from the segmental aneuploidy formation in *C. albicans*, none of the characterized segmental aneuploid regions observed in this study are flanked by inverted long-repeat sequences. Instead, many segmental aneuploid regions are flanked by T1 or T3/CNIRT4 DNA transposons. In *C. neoformans*, movements of both DNA transposons have been observed at elevated rates *in vivo* and at host temperature and contribute to development of resistance against multiple antifungal drugs [52]. It is likely that transposons play a role in segmental aneuploidy formation and contribute to chromosome instability. Based on detection of discordant read-pairs at the borders of the segmental aneuploid regions, there appears to be two likely modes of segmental aneuploidy formation. In the *MAT***a** *ura5Δ NURAT/NAT*-1 and -2 strains, forward and reverse reads of a subset of the aligned read-pairs were found on different chromosomes, and one of the regions always contained a centromere, suggesting segmental aneuploidy formation may occur in conjunction with chromosomal fusion and neo-chromosome formation. This hypothesis was confirmed by binding of a chromoblot probe for the *NAT* cassette to two distinct chromosomes. On the other hand, in the *MAT***a** *ura5Δ NURAT/NAT*-3 and *MAT*α *ura5Δ NURAT/NAT*-1 strains, segmental aneuploidy formed on the same chromosome and involved tandem duplications and inversions, likely through breakage-

fusion-bridge cycles similar to what has been observed in *C. albicans*. However, chromoblot analysis for the segmental aneuploid regions only confirmed the size increase in Chr 6 for the *MATα ura5Δ NURAT/NAT*-1 strain. Surprisingly, chromoblot analysis for both the *URA5* gene and *NAT* cassette detected the presence of a smaller chromosome in the *MAT***a** *ura5Δ NURAT/NAT*-3 and the *MATα ura5Δ NURAT/NAT*-1 strains, suggesting additional segmental aneuploidy formation mechanisms that remain to be elucidated.

Recent experimental selection experiments have shown that breakage or deletion of centromeres in *Cryptococcus* species can induce chromosomal translocation and chromosomal fusion events, which may lead to reproductive isolation, underlying the involvement of karyotypic variation in speciation during evolution [65,66]. However, detection of chromosomal instability events is extremely difficult as naturally occurring karyotypic changes are rare events derived from defective mitosis and that abnormal karyotypes often result in cellular fitness costs [67,68]. Here we showed that the *NURAT* cassette has tremendous potential for isolating cells with altered karyotypes, which may prove to be a valuable tool for elucidating mechanisms underlying chromosome instability.

Utilizing the *NURAT* cassette, we characterized the involvement of cell cycle-regulating genes in diploidization during unisexual reproduction. However, because diploidization could be detected at a low frequency during the early stages of unisexual reproduction, double selection of the *NURAT* reporter yielded mostly aneuploid isolates. Deletion of cyclin and CDK regulator genes and altered expression of *CLB3* all impacted *NURAT* recombination, suggesting perturbation of cell cycle progression suppresses homologous recombination. Because the *NURAT* reporter requires homologous recombination to detect ploidy transition events, defects in this function will likely limit the use of *NURAT* in probing ploidy transition. Despite this limitation, deletion of *PCL9* or *CKS1* did not block double selection for the *NURAT* and the recombined *NAT* cassettes. Interestingly, deletion of *PCL2*, *PCL6*, and *CKS2* almost completely prohibited double selection for the ploidy reporter, while these three genes had a minimal impact on diploid blastospore formation, suggesting diploidization and aneuploidy formation may require distinct cell cycle regulatory circuitries.

Detection of the segmental aneuploidy events in this study illuminate yet another example of the diverse mechanisms of genome plasticity in *C. neoformans*. Unlike diploidization during unisexual reproduction and polyploidization during titan cell formation in the host environment, segmental aneuploidy is more likely to be the result of a rare mitotic error than to be regulated by defined genetic pathways. Despite the rarity, three out of the five characterized segmental aneuploid isolates exhibited resistance to the antifungal azole drug fluconazole compared to their progenitor strain, which was likely due to an increase in the copy number of the azole efflux pump gene *AFR1*, suggesting that, under selection, these rare events could provide fitness benefits for these cells to adapt to environmental stresses [51]. However, the prevalence and biological significance of segmental aneuploidy in different environmental and clinical isolates or strains with different ploidy status requires further experimental exploration.

## Materials and methods

### Strains, media, and growth conditions

Strains and plasmids used in this study are listed in S5 Table. All strains were generated in the congenic *MAT***a** and *MATα* XL280 strain backgrounds [69]. Strains were frozen at -80°C in glycerol and maintained on Yeast Extract Peptone Dextrose or Glucose (YPD or YPG) agar medium for routine use. Strains harboring dominant selectable markers were grow on YPD or YPG medium supplemented with 100 μg/mL nourseothricin (NAT), 200 μg/mL G418 (NEO), or 200 μg/mL hygromycin (HYG). Synthetic dextrose or galactose medium without uracil

(SD-URA or SG-URA) was used to select uracil prototrophic progeny. Unisexual and bisexual mating assays were induced on either 5% V8 juice agar medium (pH = 7) or Murashige and Skoog (MS) medium minus sucrose (Sigma-Aldrich) in the dark at room temperature for the designated time period.

### Identification of putative cell cycle genes

To identify genes involved in cell cycle control, the key word cyclin was used to search against the *C. deneoformans* JEC21 genome on FungiDB (www.fungidb.org) [38]. BLASTP searches were performed for candidate cell cycle regulating genes against the *S. cerevisiae* genome database (www.yeastgenome.org), and then reciprocal BLASTP searches of top candidate genes in *S. cerevisiae* were conducted against the *C. deneoformans* JEC21 genome database to provide putative gene names and predicted protein functions (S1 Table).

### Expression levels of the putative cell cycle genes during unisexual reproduction

The wild-type XL280α strain was grown overnight in YPD liquid medium and adjusted to $OD_{600} = 2$ in sterile $H_2O$. Then 10 μl of cells were spot inoculated on V8 (mating-inducing condition) and YPD (non-mating condition) agar media and incubated for 36 hours. RNA extraction and qRT-PCR were performed as previously described [18]. Gene expression levels were normalized using the endogenous reference gene *GPD1* and determined by using the comparative ΔΔCt method. Expression fold change on V8 versus YPD agar media for each putative cell cycle-regulating gene was compared to *KAR5*, which has been previously shown to be expressed in the XL280α strain at a comparable level on V8 and YPD agar [18]. Primers used for qRT-PCR are listed in S6 Table.

### Deletion of putative cell cycle genes and conditional expression of *CLB3*

The primers used in this section are listed in S6 Table. Coding sequences (CDS) for six differentially expressed putative cell cycle genes *PCL2*, *PCL6*, *PCL9*, *CLB3*, *CKS1*, and *CKS2* were replaced by the dominant selectable marker *NEO* cassette through homologous recombination as previously described [70]. In brief, for each gene, a deletion construct consisting of 1 kb upstream and 1 kb downstream sequences flanking the CDS and the *NEO* cassette was generated by overlap PCR, and then the deletion construct was introduced into the wild type XL280α strain via biolistic transformation. Stable transformants were selected on YPD medium supplemented with G418 (200 mg/l) and gene replacements were confirmed by PCR. Two independent deletion mutants were generated for *PCL2*, *PCL6*, *PCL9*, *CKS1*, and *CKS2* deletion mutants.

Biolistic transformation using the deletion construct for *CLB3* failed to generate a deletion mutant. To study *CLB3*, a conditional expression allele of *CLB3* under a galactose inducible promoter was generated by replacing a 300-bp region upstream of the *CLB3* start codon with a *NEO* cassette followed by a 1034-bp promoter sequence for the *GAL7* (CNM00600) ($P_{GAL7}$) gene using the TRACE method [55,71]. To generate the regulated expression construct, the *NEO* cassette, the 1034-bp $P_{GAL7}$ sequence, the 1117-bp upstream and 1019 bp downstream sequences of the 300-bp region were cloned into the pXL1 plasmid using the Gibson cloning method resulting in plasmid pSH5 [72]. Then the regulated expression construct was PCR amplified from the plasmid pSH5 using primer pair JOHE45301/JOHE46452. For the sgRNA expression construct, the U6 promoter and the sgRNA scaffold that share 20 bp of overlapping sequence targeting the 300 bp region were amplified from XL280α genomic DNA and the plasmid pYF515 respectively, and then the intact sgRNA expression construct was generated by

overlap PCR [55,73]. The *CAS9* expression construct was amplified from the plasmid pXL1-CAS9-HYG [55]. The regulated expression construct, the sgRNA expression construct, and the *CAS9* expression construct were transformed into wild-type XL280α cells through electroporation using a BIO-RAD Gene Pulser. Stable transformants were selected on YPG medium supplemented with G418 and the correct integration of the *GAL7* promoter was confirmed in the transformant CF1715 by PCR.

To validate that the *GAL7* promoter could regulate *CLB3* expression, wild-type XL280α and CF1715 (*NEO-P $_{GAL7}$-CLB3*) strains were grown overnight in liquid YPD medium and adjusted to $OD_{600}$ = 2 in sterile $H_2O$. Then 10 μl of cells were spot-inoculated on YPD, YPG, V8, V8 + 2% glucose, and V8+ 2% galactose agar media and incubated for 36 hours. RNA extraction and qRT-PCR were performed as previously described to determine the expression level of *CLB3* [18].

## Microscopy

To test whether the putative cell cycle regulating genes contribute to sexual reproduction, wild type XL280α, two independent deletion mutants for *PCL2*, *PCL6*, *PCL9*, *CKS1*, and *CKS2*, and the conditional expression strain for *CLB3* were spot-inoculated on MS agar medium and incubated for up to three weeks for unisexual reproduction; and *MAT**a*** and *MAT*α cells of the wild type XL280 and deletion mutants for *PCL2*, *PCL6*, *PCL9*, *CKS1*, and *CKS2* were equally mixed and spot-inoculated on MS agar medium and incubated up to two weeks for bisexual reproduction. Hyphal growth on the edge of mating patches, basidia, and spore chains were captured at specified time points using a Nikon Eclipse E400 microscope equipped with a Nikon DXM1200F camera.

To observe yeast cell morphology, wild-type XL280α, two independent deletion mutants for *PCL2*, *PCL6*, *PCL9*, *CKS1*, and *CKS2*, and the conditional expression strain for *CLB3* were grown overnight in liquid YPD or YPG medium. Yeast cells were then fixed in 3.7% formaldehyde, membrane permeabilized in 1% Triton X100, and stained with 1 μg/ml DAPI (Thermo Fisher) and 1 μg/ml calcofluor white (CFW) (Sigma). Stained yeast cells were imaged using a ZEISS Imager widefield fluorescence microscope and images were processed using the software FIJI.

## Basidiospore and blastospore dissection

Dissections of basidiospores and blastospores were performed using a fiber optic needle spore dissecting system as previously described [18,74]. To obtain meiotic progeny, mating patches were inoculated on MS medium and incubated in the dark at room temperature for 1–2 weeks to allow basidiospore chain formation. Basidiospores were transferred onto YPD agar medium (YPG was used when strains expressing *CLB3* under a galactose-inducible promoter were involved), and individual basidiospores were separated. To dissect blastospores, mating patches were prepared similarly but incubated for 3–4 weeks or longer until hyphae grew further away from yeast cells on the agar surface, then the agar block containing the entire mating patch was excised and transferred to a YPD or YPG agar medium plate with an equivalent size of agar removed to fit the mating patch agar block, and nascent blastospores produced along the growing hyphae were separated onto fresh YPD or YPG medium.

## Flow cytometry

To determine ploidy, actively growing cells on solid agar medium were collected, fixed in ethanol, stained with propidium iodide, and analyzed by Fluorescence Activated Cell Sorting (FACS) using a BD FACSCanto II analyzer as previously described [18,75]. XL280α and

MN142.6 (XL280α/α *ura5*Δ::*NAT*/*ura5*Δ::*NEO*) were used as haploid and diploid controls, respectively. All FACS data were analyzed in FlowJo.

To determine whether the putative cell cycle regulating genes contributed to cell cycle progression, wild-type and deletion mutant cells were treated with hydroxyurea or nocodazole to arrest cells at G1/S and G2/M, respectively [44,45]. For G1/S arrest, cells were grown in YPD liquid medium overnight, washed in $H_2O$, readjusted to $OD_{600}$ = 0.2 in 2 ml fresh YPD liquid medium, regrown at 30˚C for 3 hours to reach exponential growth, and then hydroxyurea was added to the growing culture at a final concentration of 90 mM. Cells were then grown for an additional 3 hours to arrest cells at G1/S phase. Half of the volume of arrested cells was collected and fixed in 70% ethanol, and the other half was fixed after cell cycle release from G1/S arrest by growing in fresh YPD liquid medium for 90 minutes. For G2/M arrest, cells were prepared in the same manner and grown in the presence of nocodazole at a final concentration of 100 nM for 5 hours, and arrested cells were fixed in 70% ethanol. Fixed cells were then stained with propidium iodide and analyzed by FACS following the method described above. For the conditional expression strain of *CLB3*, the experiment was repeated in both YPD and YPG liquid media.

To analyze population ploidy dynamics during mating, the wild-type and deletion mutants were grown on YPD and V8 agar media for 24 hours and cell ploidy was determined by FACS. The *CLB3* conditional expression strain and wild type were grown on YPD, YPG, V8, V8 + 2% glucose, and V8 + 2% galactose agar media for 24 hours.

## Generation of the ploidy sensor *NURAT*

The ploidy sensor plasmid pNURAT was generated using the Gibson cloning method [72]. First, the truncated 5' and 3' *NAT* cassette sequences sharing 530 bp of the *NAT* CDS were PCR amplified from the plasmid pAI3 using primer pairs JOHE40975/JOHE41548 and JOHE41547/JOHE40976, the *URA5* expression cassette was amplified from XL280α genomic DNA using the primer pair JOHE41549/JOHE41550, and the plasmid backbone was amplified from plasmid pAI3 using the primer pair JOHE41352/JOHE41353. These PCR products share 20 bp overlapping sequences and were assembled together to generate pNURAT where the *URA5* expression cassette was inserted between the truncated 5' and 3' *NAT* cassette sequences. To insert the ploidy sensor into the genome, the safe haven locus was identified on Chr 1 in *C. deneoformans* using the identified safe haven locus in *C. neoformans*, and the plasmid pCF3 (*SH-NEO*) targeting the *C. deneoformans* safe haven locus was generated by swapping the *C. neoformans* sequence and the *NAT* cassette in pSDMA25 with the *C. deneoformans* sequences and the *NEO* cassette [50]. The *NURAT* construct was then PCR amplified from pNURAT and inserted into pCF3 to yield pCF7 (*SH-NURAT-NEO*) using the Gibson method. pCF7 was linearized with PacI and introduced into XL280α via biolistic transformation. Insertion of *NURAT-NEO* at the safe haven locus was verified in the resulting transformant CF1300 by junction PCRs and southern blot.

To generate the deletion construct for the endogenous *URA5* gene, the *HYG* cassette was PCR amplified from the plasmid pAG32 and inserted between 5' and 3' sequences flanking the *URA5* CDS using overlap PCR. The deletion construct was then introduced into XL280**a** via biolistic transformation, and replacement of the *URA5* CDS by the *HYG* cassette in the resulting transformant CF1321 was verified by junction and spanning PCRs.

To generate strains carrying both *SH-NURAT-NEO* and *ura5*Δ::*HYG*, CF1300 (XL280α *SH-NURAT-NEO*) was crossed with CF1321 (XL280**a** *ura5*Δ::*HYG*), and basidiospores were dissected following methods described above. Progeny were streaked on YPD+NAT, YPD +NEO, YPD+HYG, and SD-URA media to check viability phenotypes on each medium.

NAT-sensitive and NEO- and HYG-resistant progeny that could grow on SD-URA medium were PCR genotyped for the mating-type locus, deletion of the *URA5* gene, and the presence of the *NURAT-NEO* construct at the safe haven locus. Two progeny CF1348 (XL280**a** *ura5*Δ::*HYG SH-NURAT-NEO*) and CF1349 (XL280α *ura5*Δ::*HYG SH-NURAT-NEO*) were verified and selected for further analyses. To generate diploid strains carrying the ploidy sensor, blastospores were dissected from CF1349 and two diploid progeny (CF1610 and CF1611 XL280 α/α *ura5*Δ::*HYG/ura5*Δ::*HYG SH-NURAT-NEO/SH-NURAT-NEO*) were obtained.

To introduce the ploidy sensor into the deletion mutant strains lacking the putative cell cycle-regulating genes, *MAT*α *pcl2*Δ::*NEO*, *MAT*α *pcl6*Δ::*NEO*, *MAT*α *pcl9*Δ::*NEO*, *MAT*α *cks1*Δ::*NEO*, and *MAT*α *cks2*Δ::*NEO* were first crossed with XL280**a** to obtain deletion mutants of each gene in the *MAT***a** background, and then *MAT***a** *pcl2*Δ::*NEO* (CF1510), *MAT***a** *pcl6*::*NEO* (CF1534), *MAT***a** *pcl9*Δ::*NEO* (CF1798), *MAT***a** *cks1*Δ::*NEO* (CF1526), and *MAT***a** *cks2*Δ::*NEO* (CF1516) were crossed with XL280α *ura5*Δ::*HYG SH-NURAT-NEO* (CF1349). Basidiospores were dissected from each cross, and NAT-sensitive and NEO- and HYG-resistant progeny that could grow on SD-URA medium were PCR genotyped for the mating-type locus, deletion of the putative cell cycle gene, deletion of the *URA5* gene, and the presence of the *NURAT-NEO* construct at the safe haven locus. For each cell cycle gene, two *MAT*α progeny with the desired genotype were obtained except for *PCL6*, where one *MAT*α and one *MAT***a** progeny were obtained. For *CLB3*, the conditional expression strain XL280α $P_{GAL7}$-*CLB3-NEO* (CF1715) was crossed with XL280**a** *ura5*Δ::*HYG SH-NURAT-NEO* (CF1348), and basidiospores were dissected on YPG agar medium. NAT-sensitive, NEO- and HYG-resistant progeny that could grow on SG-URA medium were genotyped for the mating-type locus, the presence of the conditional expression construct for *CLB3*, deletion of the *URA5* gene, and the presence of the *NURAT-NEO* construct at the safe haven locus. One progeny with the desired genotype was obtained (CF1835).

## Detection of ploidy transition events using the ploidy sensor *NURAT*

To test whether the ploidy sensor *NURAT* could detect ploidy transition events, fluctuation assays were performed using haploid and diploid wild-type strains carrying the ploidy sensor. Overnight cultures for CF1348, CF1349, CF1610, and CF1611 were washed once and adjusted to $OD_{600}$ = 5 in sterile $H_2O$, and 100 μl of cells were spot inoculated on YPD or V8 pH = 7.0 agar medium and incubated in the dark at room temperature for 36 or 60 hours. After incubation, cells were collected, suspended in 300 μl sterile $H_2O$, and serially diluted by 10-fold seven times. 200 μl of cells from each of the last two serial dilutions were plated on YPD agar medium to estimate the number of colony forming units (CFUs), and 200 μl of the undiluted and the 10-fold diluted cell suspensions were plated on YPD agar medium supplemented with NAT to select for progeny with the recombined *NURAT* construct. *NURAT* recombination events per million CFU was used to determine the recombination frequency. Nourseothricin-resistant (NAT$^R$) colonies were then replica plated onto SD-URA medium to select NAT$^R$ colonies that were uracil prototrophic (Ura$^+$). The percentages of Ura$^+$ colonies among NAT$^R$ colonies were calculated to determine the double selection efficiency of the *NURAT* and *NAT* genetic constructs. For each experiment, up to eight colonies were tested for ploidy by FACS analyses. For each strain, five biological replicates were performed for each condition.

To study whether the identified putative cell cycle genes impact ploidy transitions during early mating, fluctuation assays were performed for *PCL2* (CF1779 and CF1780), *PCL6* (CF1773 and CF1774), *PCL9* (CF1806 and CF1807), *CKS1* (CF1784 and CF1787), and *CKS2* (CF1770 and 1772) genetic deletion mutants following the method above by incubating cells on V8 pH = 7.0 agar medium for 60 hours. For *CLB3* expression, cells of the wild type

(CF1349) and the conditional expression strain for *CLB3* (CF1835) were incubated on V8, V8 2% glucose, and V8 2% galactose agar media for 60 hours.

## DNA preparation, Nanopore sequencing and assembly of *C. deneoformans* XL280

The DNA for Nanopore sequencing of the XL280α genome was isolated as described previously [66]. The DNA was enriched for high molecular weight, and purified DNA was tested for its quality using NanoDrop. The samples were sequenced on the MinION system using an R9.4.1 Flow-Cell and the SQK-LSK109 library preparation kit. Nanopore sequencing was performed at the default voltage for 48 hours as per the MinION sequencing protocol provided by the manufacturer. MinION sequencing protocol and setup was controlled using the Min-KNOW software. Base-calling was performed with Guppy v4.2.2 using the parameters: config dna_r9.4.1_450bps_fast.cf—gscore_filtering, and the sequence reads obtained were used for genome assembly.

Canu v2.0 [76] was used to assemble the genome of XL280α using reads that were longer than 10 kb (-minReadLength = 10000), which yielded an estimated genome size of 19.4 Mb. The genome assembly was checked for integrity by mapping the Canu-corrected reads back to the genome assembly using minimap2 v2.14 and duplicated small contigs were discarded. These steps resulted in the generation of a chromosome-level genome assembly consisting of 14 nuclear contigs plus the mitochondrial genome. The genome assembly was then error-corrected via one round of Nanopolish v0.13.2 (using nanopore reads; https://github.com/jts/nanopolish) and five rounds of Pilon v1.23 polishing (using Illumina reads; https://github.com/broadinstitute/pilon) [77]. After the polishing, telomere sequences were identified in each chromosome, and any additional sequences flanking the telomere ends were trimmed after validation by Nanopore read-mapping. The chromosomes were numbered based on their synteny with the JEC21 genome [78]. Repetitive DNA content, including transposable elements, was analyzed with RepeatMaster version open-4.0.7 (using RepBase-20170127 and Dfam Consensus-20170127). Centromeres were predicted by detection of centromere-associated LTR elements previously reported in *C. neoformans* (Tcn1 to Tcn6) [79], and further refined by mapping onto the XL280 assembly using the position of each of the centromere flanking genes previously identified in *C. deneoformans* [80], using BLAST analyses. Both nanopore and Illumina data for the XL280 genome have been deposited at the NCBI under the accession number PRJNA720102.

## Illumina genome sequencing and read coverage assessment

To understand the nature of double selection of the *NURAT* construct and *NAT* marker in non-diploidization events, whole-genome Illumina sequencing was performed for parental strains CF1300, CF1321, CF1348 and CF1349, and five NAT[R], Ura[+] progeny (CF1354, CF1355, CF1356, CF1357, and CF1358) obtained through mitotic passaging on YPD agar medium. Genomic DNA was extracted following method as previously described [81]. Short-read library preparation and genome sequencing were conducted at the University of North Carolina at Chapel Hill's Next Generation Sequencing Facility. Paired 151-base reads were generated in an Illumina Hiseq2500 system.

To detected aneuploidy events (including segmental or whole-chromosome aneuploidy), Illumina paired-end reads were filtered with the default parameters of Trimmomatic v0.36 [82], and subsequently mapped to the *C. deneoformans* XL280 reference genome using the BWA-MEM short-read aligner (v0.7.17-r1188) with default settings [83]. Picard tools, integrated in the Genome Analysis Toolkit (GATK) v4.0.1.2 [84], was used to sort the resulting

files by coordinate, to fix read groups (modules: SORT_ORDER = coordinate; 'AddOrRepla-ceReadGroups') and to mark duplicates. Aneuploidy events were inferred from read counts calculated in 1-kb non-overlapping windows across the genome using the module "count_dna" from the Alfred package (v0.1.7) (https://github.com/tobiasrausch/alfred). These counts were subjected to median normalization and log2 transformation and the data was converted into a tiled data file (.tdf) using "igvtools toTDF" and plotted as a heatmap in IGV viewer v2.8.0 [85]. Structural events including inversions, duplications, and translocations/fusions were inferred based on the manual inspection of discordant read pairs, with LL/RR reads implying inversions and RL reads implying duplications with respect to the reference. These sets of reads are represented in IGV with different color codes after grouping and color alignments by insert size and pair orientation.

### Stress response phenotype

To test whether segmental aneuploidy conferred phenotypic variance to heat stress and the antifungal drug fluconazole, haploid and diploid wild-type strains and the five mitotically passaged double selection progeny (CF1354, CF1355, CF1356, CF1357, and Cf1358) were cultured overnight at 30˚C in liquid YPD medium. The cells were subsequently washed once with water, adjusted to OD600 = 0.8, 10-fold serially diluted, and spot inoculated on YPD and YPD agar medium supplemented with 8 μg/ml fluconazole. YPD plates were incubated at 30˚C and 37˚C, and the fluconazole plates were incubated at 30˚C for 48 to 72 hours [86].

### Pulsed-field gel electrophoresis (PFGE) and chromoblot analysis

PFGE and chromoblot analyses were performed as previously described [87]. CHEF gels were run using 1% agarose gel in 0.5X TBE at 14˚C for 96 hours with a ramped switching time from 260 seconds to 560 seconds at a voltage of 3V/cm. To separate smaller chromosomes, CHEF gels were run for 40 hours with a ramped switching time from 50 to 76 seconds at a voltage of 5 V/cm. For chromoblot analyses, probes were designed to hybridize to *URA5*, the *NAT* cassette, and the segmental aneuploid regions on Chrs 2, 6, and 13. Primers used to PCR amplify the probes were listed in S6 Table.

### Statistical analyses

Graph preparation and statistical analyses were performed using the Graphpad Prism 8 program. Student's t-test was performed for each pairwise comparison. *p* values lower than 0.05 were considered statistically significant (* indicates $0.01 < p \leq 0.05$, ** indicates $0.001 < p \leq 0.01$, *** indicates $0.0001 < p \leq 0.001$, and **** indicates $p \leq 0.0001$).

### Supporting information

**S1 Fig. Expression of putative cyclin dependent kinases and expression profiles of the putative cell cycle regulators.** (A) Differential expression patterns of genes encoding putative cyclin dependent kinases in wild-type XL280α cells incubated for 36 hours on mating-inducing V8 agar medium versus nutrient-rich YPD agar medium were examined by qRT-PCR. (B) Relative expression levels for the six differentially expressed cell cycle regulators were extrapolated from a time-course transcriptional profiling study of the wild-type strain XL280α during unisexual reproduction [39]. Expression levels on YPD medium after incubation for 12h and V8 agar medium after incubation for 12h, 24h, and 48h were plotted for these putative cell cycle genes. A black dashed line was drawn to indicate the time point assayed for these genes

in this study.
(TIF)

**S2 Fig. Expression of *CLB3* under the galactose-inducible promoter $P_{GAL7}$ and bisexual mating phenotypes of the cell cycle regulating gene deletion mutants.** (A) *CLB3* was expressed under the control of galactose-inducible promoter $P_{GAL7}$. Compared to the expression level of the wild type on YPD agar medium, $P_{GAL7}$-*CLB3* was upregulated 5.2- and 5.8-fold on YPG and V8 galactose agar media and downregulated 38.8-, 15.4-, 10.9-, and 9.9-fold on YPD, MS, V8, and V8 glucose agar media, respectively. The error bars represent the standard deviation of the mean for three biological replicates. (B) *MAT**a*** and *MATα* cells of wild type XL280 and deletion mutants for *PCL2*, *PCL6*, *PCL9*, *CKS1*, and *CKS2* were equally mixed and inoculated on MS medium to assess bisexual hyphal growth and spore formation. Hyphal growth on the edge of each colony was imaged after three days and the scale bar represents 200 μm. Spore formation was imaged after eleven days and the scale bar represents 50 μm.
(TIF)

**S3 Fig. Deletion of *CKS1* and repressed expression of *CLB3* result in pseudo-hyphal growth.** Wild type, deletion mutants of *PCL2*, *PCL6*, *PCL9*, *CKS1*, and *CKS2* were grown in liquid YPD overnight, and the conditional expression strain for *CLB3* was grown in liquid YPD and YPG medium overnight. Cells were stained with Calcofluor white and DAPI. The scale bar represents 10 μm.
(TIF)

**S4 Fig. Deletion of *CKS1* and reduced expression of *CLB3* arrest cells at G2 phase.** Overnight culture in YPD for the wild type and deletion mutant strains of *PCL2*, *PCL6*, *PCL9*, *CKS1*, and *CKS2*, and overnight culture in YPD for the conditional expression strain for *CLB3* were arrested by hydroxyurea and nocodazole to assess whether these genes regulate cell cycle progression. Cells were arrested in G1 by hydroxyurea and released to S/G2 after removal of hydroxyurea. Nocodazole arrested cells at S/G2 phase. Ploidy for the cell populations were determined by FACS.
(TIF)

**S5 Fig. Population ploidy distribution is similar between mating-inducing and -suppressing conditions.** Wild type and deletion mutants of *PCL2*, *PCL6*, *PCL9*, *CKS1*, and *CKS2* were grown on YPD and V8 agar media for 24 hours.
(TIF)

**S6 Fig. Schematic diagram for the generation of strains carrying the *NURAT* ploidy sensor.** A *MATα NURAT* strain (CF1300 XL280α *SH-NURAT-NEO*) was crossed with a *MAT**a*** *ura5Δ* strain (CF1321 XL280**a** *ura5Δ::HYG*) to generate *MAT**a*** and *MATα ura5Δ NURAT* strains (CF1348 and CF1349). The *MAT**a*** *ura5Δ NURAT/NAT*-1, -2, -3, and -4 strains (CF1354, CF1355, CF1356, and CF1347) and the *MATα ura5Δ NURAT/NAT*-1 strain (CF1358) were generated through mitotic passages of CF1348 and CF1349, respectively. All above nine strains were subjected to Illumina whole-genome sequencing. Diploid *MATα ura5Δ NURAT/NURAT* strains (CF1610 and CF1611 α/α *ura5Δ/ura5Δ NURAT/NURAT*) were generated by dissecting blastospores from CF1349. Ploidy of all strains were confirmed by FACS.
(TIF)

**S7 Fig. Schematic diagram for the ploidy transition detection assays using the *NURAT* ploidy sensor.** Overnight cultures of strains carrying the *NURAT* construct were washed and inoculated on V8 or YPD medium for the designated time period. Cells were then plated on

YPD medium supplemented with nourseothricin to select for cells with a recombined, functional *NAT* construct. Colonies derived from these cells were replica plated onto SD-URA medium to screen for NAT$^R$ cells that retained an intact *NURAT* construct. NAT$^R$, Ura$^+$ colonies were then tested for ploidy by FACS.
(TIF)

**S8 Fig. Ploidy sensor reports ploidy transition events during both mitotic and meiotic growth.** Frequencies of *NURAT* recombination in haploid strains *MAT*a *NURAT* (CF1348, only overnight culture was tested) and *MAT*α *NURAT* (CF1349), and diploid strains *MAT* α *NURAT/NURAT*-1 and *MAT*α *NURAT/NURAT*-2 (CF1610 and CF1611) grown (A) as overnight cultures in liquid YPD medium and (B) on V8 or YPD agar medium for 36 hours (scatter dot plots). NAT$^R$ colonies were replica-plated onto SD-URA medium to obtain NAT$^R$, Ura$^+$ colonies (bar graphs), and ploidy for these colonies was assessed by FACS (violin plots). Mean values of five independent experiments were plotted for the *NURAT* recombination frequencies and the Ura$^+$ percentiles among NAT$^R$ colonies; error bars represent standard deviations. Student's T-test was performed for each pairwise comparison. $^{**}$ indicates $0.001 < p \leq 0.01$.
(TIF)

**S9 Fig. Flanking sequences of segmented regions show distinct modes of segmental aneuploid formation.** For each *NURAT/NAT* progeny, sequencing reads at the borders of segmentally duplicated regions were analyzed. Blue, red, and green bubbles indicate forward and reverse reads that were aligned to two different chromosomal positions. Sequence alignments of these reads were shown in the panels on the right of the chromosome diagrams. Chimeric reads aligning to two different chromosomal positions were highlighted in connected boxes. Sequencing reads aligned to segmentally duplicated regions from three chromosomes were identified in the *MAT*a *ura5*Δ *NURAT/NAT*-1, -2, and -4 strains, suggesting fusion of these regions. T1 and T3/CNIRT4 transposable element movements were detected flanking some of the regions in the *MAT*a *ura5*Δ *NURAT/NAT*-1, -2, and -3 strains and the *MAT*α *ura5*Δ *NURAT/NAT*-1 strain. In the *MAT*α *ura5*Δ *NURAT/NAT*-1 strain, tandem duplication and inversion events were detected in the segmentally duplicated regions.
(TIF)

**S10 Fig. Karyotypic changes are associated with segmental aneuploid formation.** CHEF gel electrophoresis separation of chromosomes was performed under different conditions to separate larger or smaller chromosomes. Karyotypic changes (highlighted in green and red arrows) were observed for strains with segmental aneuploidy (*MAT*a *ura5*Δ *NAT/NURAT*-1, -2, -3, -4 and *MAT*α *ura5*Δ *NAT/NURAT*-1) compared with wild type and parental strains. Chromoblot analyses with probes recognizing (A) *URA5* and *NAT*, and segmental aneuploid portions of (B) Chrs 2 and 6 and (C) Chr 13 confirmed the karyotypic changes. Strains are highlighted in red when the probed sequences are within segmental aneuploid regions.
(TIF)

**S11 Fig. *PCL9* and *CLB3* contribute to aneuploidy formation.** Individual colonies with ploidies above 1.6 were identified as outliers and removed from Fig 5 and data were replotted for (A) *pcl9*Δ and *cks1*Δ mutants and (B) P$_{GAL7}$-*CLB3*. Student's t-tests with Bonferroni correction for 4 and 9 repeated tests were performed for each pairwise comparison for panel A and panel B, respectively. *p* value lower than 0.0125 (A) or 0.0056 (B) was considered statistically significant (*).
(TIF)

**S1 Table. Identified putative cell cycle regulators in *Cryptococcus deneoforman*.**
(XLSX)

**S2 Table. Deletion of cell cycle regulator genes impacted diploid blastospore formation.**
(DOCX)

**S3 Table. Mitotically passaged yeast cells maintained stable ploidy.**
(DOCX)

**S4 Table. G1/S- and G2/M-phase population distribution profiles.** Wild type and cell cycle regulator mutants were arrested with hydroxyurea and nocodazole and population distributions were determined by FACS.
(DOCX)

**S5 Table. Strains and plasmids used in this study.**
(DOCX)

**S6 Table. Primers used in this study.**
(DOCX)

**S1 Data. Underlying numerical data for all graphs and summary statistics.**
(XLSX)

## Acknowledgments

We thank Shelby Priest for critical reading of the manuscript and thank the helpful suggestions and discussions from members of the Heitman lab.

## Author Contributions

**Conceptualization:** Ci Fu, Joseph Heitman.

**Data curation:** Ci Fu, Marco A. Coelho.

**Formal analysis:** Ci Fu, Aaliyah Davy, Simeon Holmes, Sheng Sun, Vikas Yadav, Asiya Gusa, Marco A. Coelho.

**Funding acquisition:** Joseph Heitman.

**Investigation:** Ci Fu, Aaliyah Davy, Simeon Holmes, Sheng Sun, Vikas Yadav, Asiya Gusa, Marco A. Coelho, Joseph Heitman.

**Methodology:** Ci Fu, Marco A. Coelho.

**Resources:** Ci Fu, Joseph Heitman.

**Supervision:** Joseph Heitman.

**Validation:** Ci Fu, Sheng Sun, Marco A. Coelho.

**Writing – original draft:** Ci Fu.

**Writing – review & editing:** Ci Fu, Sheng Sun, Vikas Yadav, Marco A. Coelho, Joseph Heitman.

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
