## [Decision Letter · Decision Letter 0]

6 Jul 2021

Dear Dr Fu,

Thank you very much for submitting your Research Article entitled 'Dynamic genome plasticity during unisexual reproduction in the human fungal pathogen Cryptococcus deneoformans' to PLOS Genetics.

The manuscript was fully evaluated at the editorial level and by independent peer reviewers. As you will see from the reviewers' comments below, there was a wide range of opinions about this study.  Given that situation, I re-read the manuscript carefully after reading the reviews.  I feel that you have a contribution of high significance, and that it includes some quite elegant analysis.  You cover a lot of ground in the study, and quite naturally this feature invited a lot of comments.  I think that all of the points raised are good ones, but I'd like to emphasize two points in particular that came up.  (They will make more sense when you read each entire comment.)

- Rev 3 stated, "The authors need to sequence multiple diploid isolates from the yeast cultures (not just the aneuploid blastospores) to determine if segmental aneuploidies are also present in these strains, and if so, whether they show similar characteristics."  This analysis would be really interesting, but it could take you in yet another direction.  I'll leave it up to you as to whether you want to sequence a few isolates or perhaps dial down your claim of specificity.

- Rev 3 stated, "The NURAT system is not well described. For an individual unfamiliar with the system, and particularly its interaction with homologous recombination, the description is very difficult to comprehend."  I agree, and in fact I struggled with analogous issues with UAU1.  Give it some thought - maybe a diagram could present a sequence of events that gives rise to each phenotype.

- Rev 2 stated, "Some further insight into the specific role of cyclins in this process would be of general interest."  Although I appreciate that you are using the cyclin mutants to dissect a process rather than to understand cycling function per se, I think that any insight you could provide about cyclin function would be helpful.  (As a related note, though not a specific reviewer comment, is it correct that you were unable to delete CLB3, and if so have you any idea about why?  You mentioned that another group has deleted the gene.)

- Rev 2 stated, "Flow cytometry gating strategy to exclude doublets must be described and a representative scatterplot included in the supplemental material."  This makes sense to me.

- Rev 1 stated, "Lines 193-198.  I found this description of the expression analysis to be somewhat confusing with regard to the description of the different behavior of PCL2. It is unclear from the description why this gene was different.  Can a bit more detail be provided to better explain the difference?"  Agreed.

Based on the reviews, we will not be able to accept this version of the manuscript, but we would be willing to review a much-revised version. We cannot, of course, promise publication at that time.  Should you decide to revise the manuscript for further consideration here, your revisions should address the specific points made by each reviewer. We will also require a detailed list of your responses to the review comments and a description of the changes you have made in the manuscript.

If you decide to revise the manuscript for further consideration at PLOS Genetics, please aim to resubmit within the next 60 days, unless it will take extra time to address the concerns of the reviewers, in which case we would appreciate an expected resubmission date by email to plosgenetics@plos.org.

[LINK]

We are sorry that we cannot be more positive about your manuscript at this stage. Please do not hesitate to contact us if you have any concerns or questions.

Yours sincerely,

Aaron P. Mitchell, PhD

Guest Editor

PLOS Genetics

Gregory P. Copenhaver

Editor-in-Chief

PLOS Genetics

Reviewer's Responses to Questions

**Comments to the Authors:**

Reviewer #1: To investigate the functions underlying ploidy change during unisexual reproduction in C. deneoformans, Fu et al. first identified cell cycle-regulating genes encoding cyclins, cyclin-dependent kinases (CDK), and CDK regulators. They subsequently identified four cyclin genes and two CDK regulator genes for focused analysis based differential expression during unisexual reproduction. In general, they identified genes that contributed to diploidization. They also developed and employed a NAT-URA construct to monitor ploidy changes during unisexual reproduction and to detect segmental aneuploidy. Overall, the results in the manuscript provide an interesting illustration of aspects of genome plasticity (including segmental aneuploidy) for an important fungal pathogen in the context of cell cycle regulating functions and unisexual reproduction. For the most part, the data are clearly presented and support the conclusions. I only have a few minor issues to point out.

Lines 193-198. I found this description of the expression analysis to be somewhat confusing with regard to the description of the different behavior of PCL2. It is unclear from the description why this gene was different. Can a bit more detail be provided to better explain the difference?

Line 275-277. Something is wrong with construction of this sentence.

Line 283. Unclear phrase “more than half of the pcl6∆ blastospores”

Line 388. …tested were found to be resistant…

Reviewer #2: Fu et al., observe the impact of cyclin and cyclin dependent kinases on ploidy during growth in rich media and mating media (V8) over time, leveraging the ability of C. deneoformans to undergo unisexual reproduction to study genetic roles during yeast-phase growth and blastospore production. The authors report increased expression of PCL9, PCL6, and CKS1 and decreased expression of PCL2 and CLB3 under mating conditions. Using single mutants, they identify CKS1 and CLB3 as having an impact on spore production and also observe that on V8, mutants were defective for blastospores, with low germination and different ploidy (haploid or diploid) compared to the parental control (diploid, rare aneuploid). Using flow cytometry and a Nat/Ura (NURAT) reporter sensitive to homologous recombination and segmental duplication/diploidy, they find that growth on V8 alters cell ploidy and recombination rates. Interestingly, they also observe high levels of aneuploidy/segmental duplication in areas of the genome relevant for drug resistance (Ch1) during yeast-phase growth, and highlight a role for their genes of interest in regulating this process.

Overall, the authors have raised the possibility that there is a mating-specific mechanism for ploidy increase that is regulated by a mating-specific suite of cyclin and CDKs in C. deneoformans. Despite these interesting findings, it remains unclear what, if any, direct role these genes are playing in regulating ploidy, aneuploidy, or homologous recombination, how these genes interact with each other, why they are differentially regulated, or their shared or divergent targets. It is already well established that cyclins impact ploidy in a variety of systems, so it’s not surprising that the cyclins in C. deneoformans will do the same.

Some further insight into the specific role of cyclins in this process would be of general interest. For example, in the discussion, the authors raise a number of hypotheses about regulation of Pho85, but do not test a pho85 mutant for impacts on genome stability or investigate its potential as a target for their cyclins of interest. Alternately, why would there be a higher requirement for Glc7-regulation by Pcl6 during V8 growth than during rich-media growth? Can the authors identify a specific Cks1 target to support their hypothesis that Cks1 and Cks2 are functionally divergent rather than just differentially expressed?

Overall, the manuscript is well written, and figures are clear and concise. There are some small points where additional detail is needed for methodology (see minor comments below).

Line 258: for CLB3, it would be better to see the S5B data brought into the main text.

This finding of V8 altering expression of cell cycle genes should be placed in the context of cell cycle progression. The authors should consider citing Kelliher et al as a key paper comparing S. cerevisae cell cycle and C. neoformans under rich media conditions. However, altered expression of these genes on different media is consistent with previous findings (which the authors point out and cite).

For statistical analysis, in some instances it appears the authors are reporting p-values using Student’s t-test without correcting for repeated testing (e.g., Figure 5A, mutants vs WT). If true, this is not appropriate. The authors must correct for multiple comparisons.

Flow cytometry gating strategy to exclude doublets must be described and a representative scatterplot included in the supplemental material.

Figure S3 DAPI staining hasn’t been particularly effective. It would be better to use a nuclear dye that fluoresces in a different wavelength from CFW so the reader can differentiate CFW staining of septa, bud scars, and inappropriate chitin deposition from nuclear material.

Line 135: these references aren’t strictly focused on C. deneoformans ploidy stimuli, but instead show the impact of specific stimuli (density, QSP, etc.) on C. neoformans. There are others that would be a better fit, including work by Zhao et al., 2020 in Current Biology.

Line 287: this is just a bit confusing. “all single colonies derived from diploid blastospores were diploid, and only one single colony derived from mixed haploid-diploid blastospores remained a mixed-ploidy population. All other single colonies from the mixed 1N/2N blastospores were either haploid or diploid”

Does this mean “all single colonies derived from diploid pcl6 blastospores were diploid, and only one single colony derived from mixed pcl6 haploid-diploid blastospores remained a mixed-ploidy population. All other single colonies from the mixed pcl6 1N/2N blastospores were either haploid or diploid”

or

“all single colonies derived from both WT and pcl6 diploid blastospores were diploid, and only one single colony derived from mixed plc6 haploid-diploid blastospores remained a mixed-ploidy population. All other single colonies from the mixed pcl6 1N/2N blastospores were either haploid or diploid”

I know it’s functionally the same thing in terms of data interpretation, but it’s just a bit of a slog for the reader and could be clarified.

Line 480: what is meant by “sufficient”? Is this supported by a power calculation, or just a matter of convenience?

The references for use of HU and nocodozole for arrest at G1/S and G2/M phase don’t actually show that these drugs promote cell cycle arrest per se. This is a bit misleading. References to these techniques in other yeast, where they are well established, would be more appropriate. To their credit, the data the authors present are the best proof I’ve seen that these tools are of use in C. deneoformans.

Reviewer #3: The manuscript by Fu and colleagues explores the 6 cyclins, cyclin-dependent kinases, and other CDK regulatory genes with differential expression during unisexual reproduction in the Cryptococcus deneoformans strain XL280 using a mutagenesis approach. The authors also develop a fate mapping system for C. deneoformans to detect homologous recombination and the presence of locus duplications in the genome associated with unisexual reproduction associated, which the authors refer to as NURAT. While the manuscript presents interesting data, and is well-written, there are a number of concerns related to data presentation and analysis, whether the NURAT system can be used to for the purposes the authors propose, and whether the conclusions drawn are appropriate given the limited experimental data presented. Detailed concerns are as follows:

Major Concerns:

1. Given that the authors analyzed the phenotypes of the mutants in unisexual mating in Figure 1, and then in the yeast form in Figures S3-S5, the authors also need to analyze the mutants for their phenotype in bisexual mating. This is particularly pertinent given that many of the mutants show defects in unisexual mating associated with hyphal growth and sporulation – both of which are also critical in bisexual mating and thus it is important to know if these cyclins are utilized in both of the mating systems or are only used during unisexual mating. This point is critical for the overall conclusions of the manuscript as the authors often make statements referring only to unisexual mating, while at other time they make broad generalizations and the reasoning for this is often not transparent.

2. The NURAT system is not well described. For an individual unfamiliar with the system, and particularly its interaction with homologous recombination, the description is very difficult to comprehend. Questions that arise are: When is homologous recombination occurring? During integration of the construct into the genome? During mitosis? During meiosis? How can you differentiate between homologous recombination occurring during mitosis and during meiosis in this system? Why wouldn’t two short versions of the NURAT construct be produced in diploids or polyploids if they undergo multiple rounds of endoreplication? Why didn’t you detect any of these?

3. While I believe I understand why the diploid NURAT strains were originally developed as a mechanism to verify the NURAT construct was working as expected, I do not understand why the authors then used these strains in Figure 3B when they were testing their experimental hypothesis. It is not apparent to me what relevance performing these studies in a diploid has to the overall focus of the studies or their conclusions. Rather, the utilization of the diploid strain may inadvertently lead the authors to draw inappropriate conclusions about ploidy using these artificial diploids. In other fungal species such as S. cerevisiae strains with artificially increased ploidy have previously been shown to have increased rates of genome modifications similar to those the authors observed.

4. Based on the red/blue designations in the legend of Figure 4 it is unclear whether the authors have sufficient read depth to determine how many copies of each of these regions are present. 2 copies? 4 copies? 10 copies?

5. Genomic analysis of segmental aneuploidy sequencing was only performed on multiple blastospore derived segmental aneuploidies, and not on any of the diploid isolates identified in the yeast cultures. Yet the authors conclude that their results are both specific to unisexual mating AND will be a generalizable trait in Cryptococcus. The authors need to sequence multiple diploid isolates from the yeast cultures (not just the aneuploid blastospores) to determine if segmental aneuploidies are also present in these strains, and if so, whether they show similar characteristics.

6. While the authors show an association of the transposons with the observed segmental aneuploidies, and therefore postulate the transposons contribute to aneuploidy formation in this region, the authors do ot provide any experimental proof to support this claim. For example, if the transposon is important for formation of the aneuploidy in this region, then deletion of the transposon should result in a reduction in aneuploidy formation at this site and/or extension of the aneuploidy to the next transposon in the genome. Given that all of the blastospore isolates sequenced amplified this region, this region seems to have a high target rate and therefore perturbations should be able to be readily identifiable.

7. Because the NURAT reporter system requires homologous recombination events that occur during mitosis, it is unclear how this reporter can be used to disentangle the impact of alterations in response to both environment (i.e. media, temperature) and mutation (i.e. cell cycle mutants) on homologous recombination vs. ploidy because the function of the NURAT system is inherently dependent upon both processes to function. Thus, a circular argument is created where the system must first be functioning normally in order to determine dysfunctions.

8. XL280 is a spontaneous mutant of the wildtype C. deneoformans strain B3501 that has enhanced unisexual mating. Did the authors explore expression of the 6 cell cycle genes in the wildtype B3501 to determine whether the enhanced unisexual mating of XL280 is due to altered expression of one or more of these genes?

Minor Concerns:

9. Figure 1A – The breakpoint on the graph is not optimal as it is at a critical point for viewing some of the data. Also, some of the error bars appear to be quite large while others are small. Use of dots to represent spread in the replicates would be more transparent and allow better interpretation of the outliers and replicates that could be influencing the data interpretation and statistical analysis.

10. Figure S4 – It is impossible to clearly visualize the 1C and 2C peaks that correspond to each of the individual samples when all the samples are presented on the same plot. Individual plots need to be shown as well as the overlay plot so that the proportion of cells in each individual sample 1C and 2C peak can be easily determined, and then the overlay can be used compare the samples to each other.

11. Figure 2 – A diagram depicting the difference between blastospores and spores collected from the basidium would be useful. Germination rate can easily be added to Figure 2 and Table 2 can then be moved to the Supplemental Data as most of the information is redundant with the figure.

12. Lines 290-91 – Clarify the hypothesis of the experiment and how the results justify the conclusion statement (i.e. because stable diploid colonies were observed Pcl6 is not required for diploid maintenance)

13. Lines 301-302 – Include a reference for UAU1

14. Figure 4C – Fluconazole resistance in C. deneoformans (or any of the other Cryptococcus species) is not defined. In addition, the most appropriate method for determining changes in susceptibility to fluconazole would be a broth dilution assay, not a spot assay, and determining the IC50 and IC90 for the strains.

15. Lines 421-424 - …some…all… except… This is a lot of caveats for a single sentence. Recommend simplifying the sentence to read: ….T1 or T3 transposon movements were detected in sequences flanking some of the NURAT/NAT progeny, suggesting that transposable elements may contribute to formation of some of the segmental aneuploidies.

16. Figure 5 and Lines 478-87 – It is unclear how much the average ploidy calculation is being driven by the outliers with high ploidy. An outlier analysis should be performed and/or analyses thaking these outliers into account either statistically or via comparison/contrast.

**Have all data underlying the figures and results presented in the manuscript been provided?**

Reviewer #1: Yes

Reviewer #2: Yes

Reviewer #3: Yes

PLOS authors have the option to publish the peer review history of their article (what does this mean?). If published, this will include your full peer review and any attached files.

Reviewer #1: No

Reviewer #2: No

Reviewer #3: No

---

## [Decision Letter · Decision Letter 1]

8 Nov 2021

Dear Dr Fu,

We are pleased to inform you that your manuscript entitled "Dynamic genome plasticity during unisexual reproduction in the human fungal pathogen Cryptococcus deneoformans" has been editorially accepted for publication in PLOS Genetics. Congratulations!

Yours sincerely,

Gregory P. Copenhaver

Editor-in-Chief

PLOS Genetics

Comments from the reviewers (if applicable):

Reviewer's Responses to Questions

**Comments to the Authors:**

Reviewer #2: All my coments have been addressed.

**Have all data underlying the figures and results presented in the manuscript been provided?**

Reviewer #2: Yes

PLOS authors have the option to publish the peer review history of their article (what does this mean?). If published, this will include your full peer review and any attached files.

Reviewer #2: No

**Data Deposition**

http://datadryad.org/submit?journalID=pgenetics&manu=PGENETICS-D-21-00761R1

**Press Queries**

---

## [Editor Report · Acceptance letter]

24 Nov 2021

PGENETICS-D-21-00761R1 

Dynamic genome plasticity during unisexual reproduction in the human fungal pathogen </i>Cryptococcus deneoformans </i> 

Dear Dr Heitman, 

We are pleased to inform you that your manuscript entitled "Dynamic genome plasticity during unisexual reproduction in the human fungal pathogen </i>Cryptococcus deneoformans </i>" has been formally accepted for publication in PLOS Genetics! Your manuscript is now with our production department and you will be notified of the publication date in due course.

With kind regards,

Katalin Szabo

PLOS Genetics

On behalf of:
